REPLICATION STUDY

# Replication Study: A coding-independent function of gene and pseudogene mRNAs regulates tumour biology

John Kerwin[1†], Israr Khan[2], Reproducibility Project: Cancer Biology[3,4]*

[1]University of Maryland, College Park, United States; [2]Alamo Laboratories Inc, San Antonio, United States; [3]Science Exchange, Palo Alto, United States; [4]Center for Open Science, Charlottesville, United States

*For correspondence:
tim@cos.io;
nicole@scienceexchange.com

Present address: †Biogen, Cambridge, United States

Group author details:
Reproducibility Project: Cancer Biology See page 16

**Abstract** As part of the Reproducibility Project: Cancer Biology we published a Registered Report (Khan et al., 2015), that described how we intended to replicate selected experiments from the paper "A coding-independent function of gene and pseudogene mRNAs regulates tumour biology" (Poliseno et al., 2010). Here we report the results. We found *PTEN* depletion in the prostate cancer cell line DU145 did not detectably impact expression of the corresponding pseudogene *PTENP1*. Similarly, depletion of *PTENP1* did not impact *PTEN* mRNA levels. The original study reported *PTEN* or *PTENP1* depletion statistically reduced the corresponding pseudogene or gene (Figure 2G; Poliseno et al., 2010). *PTEN* and/or *PTENP1* depletion in DU145 cells decreased PTEN protein expression, which was similar to the original study (Figure 2H; Poliseno et al., 2010). Further, depletion of *PTEN* and/or *PTENP1* increased DU145 proliferation compared to non-targeting siRNA, which was in the same direction as the original study (Figure 2F; Poliseno et al., 2010), but not statistically significant. We found *PTEN* 3'UTR overexpression in DU145 cells did not impact *PTENP1* expression, while the original study reported *PTEN* 3'UTR increased *PTENP1* levels (Figure 4A; Poliseno et al., 2010). Overexpression of *PTEN* 3'UTR also statistically decreased DU145 proliferation compared to controls, which was similar to the findings reported in the original study (Figure 4A; Poliseno et al., 2010). Differences between the original study and this replication attempt, such as level of knockdown efficiency and cellular confluence, are factors that might have influenced the results. Finally, where possible, we report meta-analyses for each result.

## Introduction

The Reproducibility Project: Cancer Biology (RP:CB) is a collaboration between the Center for Open Science and Science Exchange that seeks to address concerns about reproducibility in scientific research by conducting replications of selected experiments from a number of high-profile papers in the field of cancer biology (*Errington et al., 2014*). For each of these papers a Registered Report detailing the proposed experimental designs and protocols for the replications was peer reviewed and published prior to data collection. The present paper is a Replication Study that reports the results of the replication experiments detailed in the Registered Report (*Khan et al., 2015*) for a paper by Poliseno et al., and uses a number of approaches to compare the outcomes of the original experiments and the replications.

In 2010, Poliseno et al. reported *PTENP1*, the pseudogene of the tumor suppressor PTEN, acted as a repressor (molecular sponge) of PTEN-targeting microRNAs and, in turn, regulated cellular PTEN expression and function. Pseudogenes are non-coding genomic DNA sequences that share high sequence similarity with their cognate protein-coding genes (*Haddadi et al., 2018*). The 3'UTR sequences of *PTEN* and *PTENP1* share common putative microRNA binding site and overexpression

of *miR-19b* and *miR-20a* in the prostate cancer cell line DU145 resulted in decreased *PTEN* and *PTENP1* mRNA levels (*Poliseno et al., 2010*). The regulatory role of *PTENP1* was demonstrated in knockdown experiments where reduction of *PTENP1* resulted in decreased *PTEN* mRNA and PTEN protein levels and increased proliferation of DU145 cells (*Poliseno et al., 2010*). Similar biological activity of the 3′UTR of *PTEN* was also reported where overexpression of *PTEN* 3′UTR derepressed *PTENP1* expression and inhibited DU145 proliferation (*Poliseno et al., 2010*).

The Registered Report for the paper by Poliseno et al. described the experiments to be replicated (Figures 1D, 2F–H and 4A), and summarized the current evidence for these findings (*Khan et al., 2015*). Since that publication additional studies have reported the biological activity of *PTENP1* in various tumors. In esophageal squamous cell carcinoma (*Gong et al., 2017*) and oral squamous cell carcinoma (*Gao et al., 2017*), overexpression of *PTENP1* decreased proliferation and colony formation *in vitro* and inhibited tumor growth in xenograft models. In head and neck squamous cell carcinoma (*Liu et al., 2017*), hepatocellular carcinoma (*Chen et al., 2015*; *Qian et al., 2017*), and bladder cancer (*Zheng et al., 2018*), *PTENP1* overexpression increased *PTEN* mRNA expression and decreased proliferation, colony formation, invasion, and migration *in vitro* and inhibited growth in xenograft models. In gastric cancer, *PTENP1* overexpression led to increased *PTEN* mRNA and PTEN protein levels, decreased cell proliferation and induced apoptosis, and inhibited migration and invasive ability of gastric cancer cells (*Zhang et al., 2017*). In clear-cell renal cell carcinoma overexpression of *PTENP1* in cells reduced cell proliferation, migration, and invasion *in vitro* and tumor growth and metastasis in xenograft models (*Yu et al., 2014*). Overexpression of *PTENP1* was reported to decrease proliferation, colony formation, and migration in the breast cancer cell line MCF7 (*Chen et al., 2017*; *Gao et al., 2019*; *Shi et al., 2018*), while *Yndestad et al., 2018* reported decreased *PTEN* mRNA expression and accelerated MCF7 tumor growth. However, in ER-negative breast cancer cells, *PTENP1* upregulation increased *PTEN* mRNA expression and inhibited tumor progression (*Gao et al., 2019*; *Li et al., 2017*; *Shi et al., 2018*; *Yndestad et al., 2018*). Additional competing endogenous RNAs (ceRNAs) that modulate *PTEN* through microRNA competition were identified using a computational model expanding upon the work reported in *Poliseno et al. (2010)* to propose a unifying hypothesis of regulatory networks composed of ceRNAs (*Tay et al., 2011*). In addition to *PTEN/PTENP1*, the pseudogene of oncogenic *BRAF*, *BRAFP1* has been reported to regulate BRAF expression and subsequently affect MAPK signaling and proliferation, and drive tumorigenesis *in vivo* (*Karreth et al., 2015*). Recently, it has also been reported an extensive gene/pseudogene network that is comprised of multiple microRNAs and multiple pseudogenes derived from a single parental gene (*Chan et al., 2018*) with the potential of thousands of novel pseudogene-gene associations beginning to be explored through bioinformatic tools that have been developed (*Johnson et al., 2019*).

The outcome measures reported in this Replication Study will be aggregated with those from the other Replication Studies to create a dataset that will be examined to provide evidence about reproducibility of cancer biology research, and to identify factors that influence reproducibility more generally.

## Results and discussion

### Cell growth assay following siRNA transfection

We independently replicated an experiment to examine the consequences of *PTEN* and/or *PTENP1* downregulation on DU145 cell proliferation. This experiment was described in Protocol 2 in the Registered Report (*Khan et al., 2015*) and is similar to Figure 2H of *Poliseno et al. (2010)* that reported *PTENP1* downregulation accelerated cell proliferation with the strongest effect observed when *PTEN* and *PTENP1* were both silenced. We utilized the same designed custom siRNA pools as the original study to specifically target either *PTEN* or *PTENP1*, as well as a commercially available *PTEN* siRNA pool that binds common sequences in *PTEN* and *PTENP1*. Proliferative activity was determined using the crystal violet assay each day after transfection for a total of 5 days with results presented relative to the values at the start of the time course for each condition (i.e., for each condition the average relative cell number at the start of the time course was set to 1) (*Figure 1*,

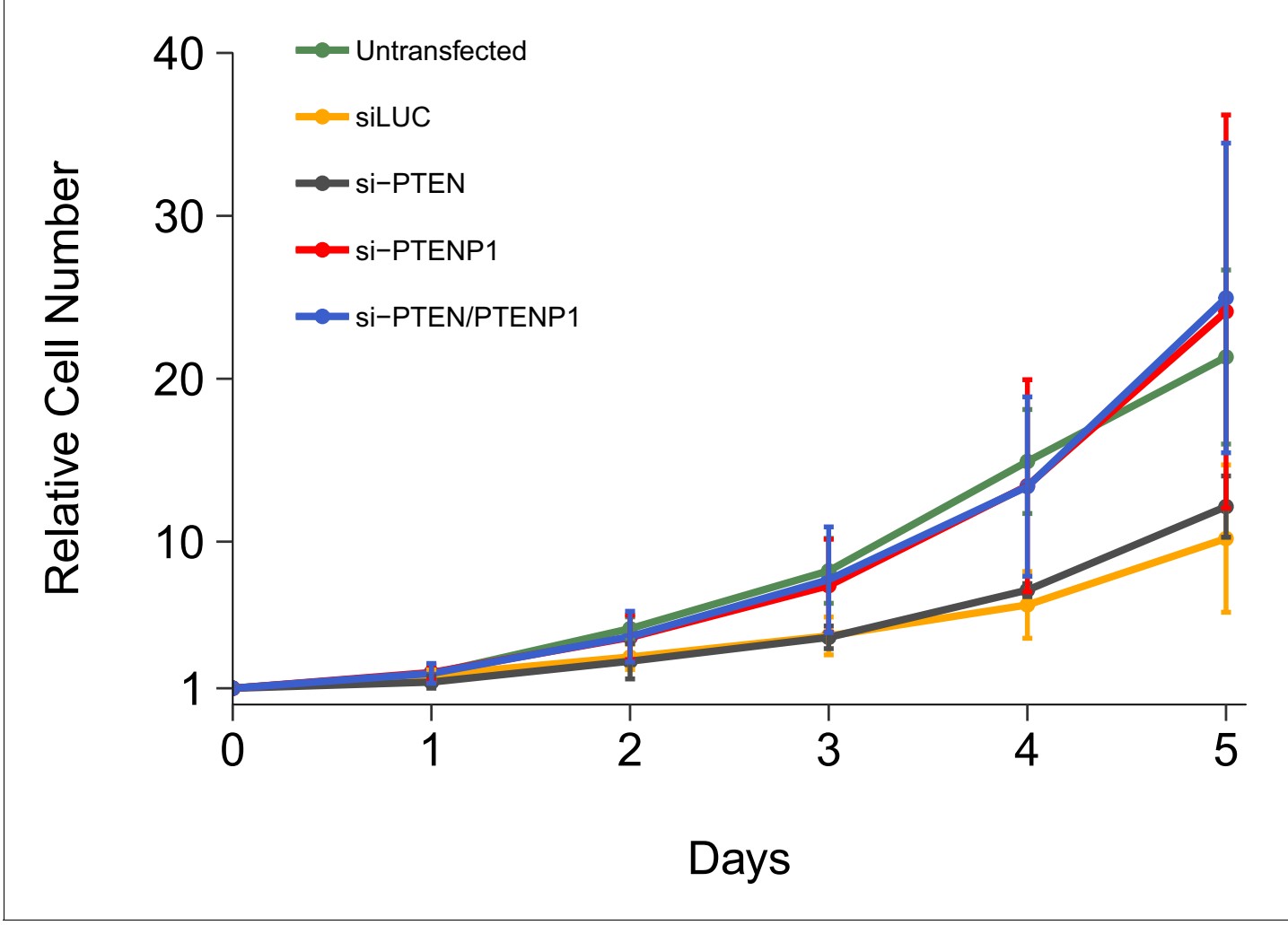

**Figure 1.** Cell growth of DU145 cells depleted of *PTEN* and/or *PTENP1*. DU145 cells were transfected with either a non-targeting siRNA (siLUC), si-PTEN, si-PTENP1, or an siRNA pool targeting *PTEN* and *PTENP1* (si-PTEN/PTENP1), or not transfected. Crystal violet proliferation assays were performed each day as indicated starting the day after transfection. Relative cell number was calculated relative to the average Day 0 values for each condition. Means reported and error bars represent *SD* from five independent biological repeats. Two-way ANOVA interaction between *PTEN* (targeted or not-targeted) and *PTENP1* (targeted or not-targeted) on Day 5 relative cell numbers: $F_{(1,16)} = 0.02$, $p=0.878$; main effect of *PTEN*: $F_{(1,16)} = 0.15$, $p=0.703$; main effect of *PTENP1*: $F_{(1,16)} = 13.8$, $p=0.0019$. Planned contrasts between siLUC and si-PTEN: $t(16) = 0.38$, uncorrected $p=0.705$ with *a priori* Bonferroni adjusted significance threshold of 0.01, Bonferroni corrected $p>0.99$; siLUC and si-PTENP1: $t(16) = 2.73$, uncorrected $p=0.015$, Bonferroni corrected $p=0.074$; siLUC and si-PTEN/PTENP1: $t(16) = 2.90$, uncorrected $p=0.011$, Bonferroni corrected $p=0.053$; si-PTEN/PTENP1 and si-PTEN: $t(16) = 2.51$, uncorrected $p=0.023$, Bonferroni corrected $p=0.115$; si-PTEN/PTENP1 and si-PTENP1: $t(16) = 0.16$, uncorrected $p=0.872$, Bonferroni corrected $p>0.99$. Additional details for this experiment can be found at https://osf.io/kjmxj/.

The online version of this article includes the following figure supplement(s) for figure 1:

**Figure supplement 1.** Alternative visualization of cell growth.

---

*Figure 1—figure supplement 1A*). We found that compared to cells transfected with non-targeting siRNA (average Day 5 relative cell number (Avg Day 5#) = 10.2), there was increased cell proliferation when cells were transfected with either the *PTENP1* specific siRNA (Avg Day 5# = 24.1) or the siRNA pool targeting *PTEN* and *PTENP1* (Avg Day 5# = 25.0). Additionally, there was a minor increase in cell proliferation when cells were transfected with the *PTEN* specific siRNA (Avg Day 5# = 12.2). Interestingly, untransfected cells, a condition not reported in the original study, displayed a cell proliferation profile (Avg Day 5# = 21.3) more similar to cells transfected with either the *PTENP1* specific siRNA or the siRNA pool targeting *PTEN* and *PTENP1* than the non-targeting siRNA. This could be a result of transfection-mediated cytotoxicity and/or off-target siRNA effects

that can generate cellular stress leading to decreased cell growth (*Antczak et al., 2014*; *Fedorov, 2006*; *Wei et al., 2012*) and is a factor to consider when interpreting these results, especially when considering the low level of increased cell growth in cells depleted of *PTEN* despite efficient knockdown (*Figure 2A*). The original study reported increased cell proliferation, compared to non-targeting siRNA, when cells were transfected with either the *PTEN* specific siRNA (Avg Day 5# = 4.15) or the *PTENP1* specific siRNA (Avg Day 5# = 2.61), with the strongest effect being observed when cells were transfected with the siRNA pool targeting *PTEN* and *PTENP1* (Avg Day 5# = 5.41) (*Poliseno et al., 2010*). In the original study the control condition (cells transfected with non-targeting siRNA) had a minimal impact on cell proliferation during the course of the experiment (Avg Day 5# = 1.60) (*Poliseno et al., 2010*). Importantly, the results of both studies should take into consideration that the doubling time for DU145 cells has been reported as ~29 hr (*Lin et al., 1998*), which would correspond to an Avg Day 5# of ~18, assuming cells maintained exponential phase growth. Additionally, the difference in achieved knockdown between the original study and this

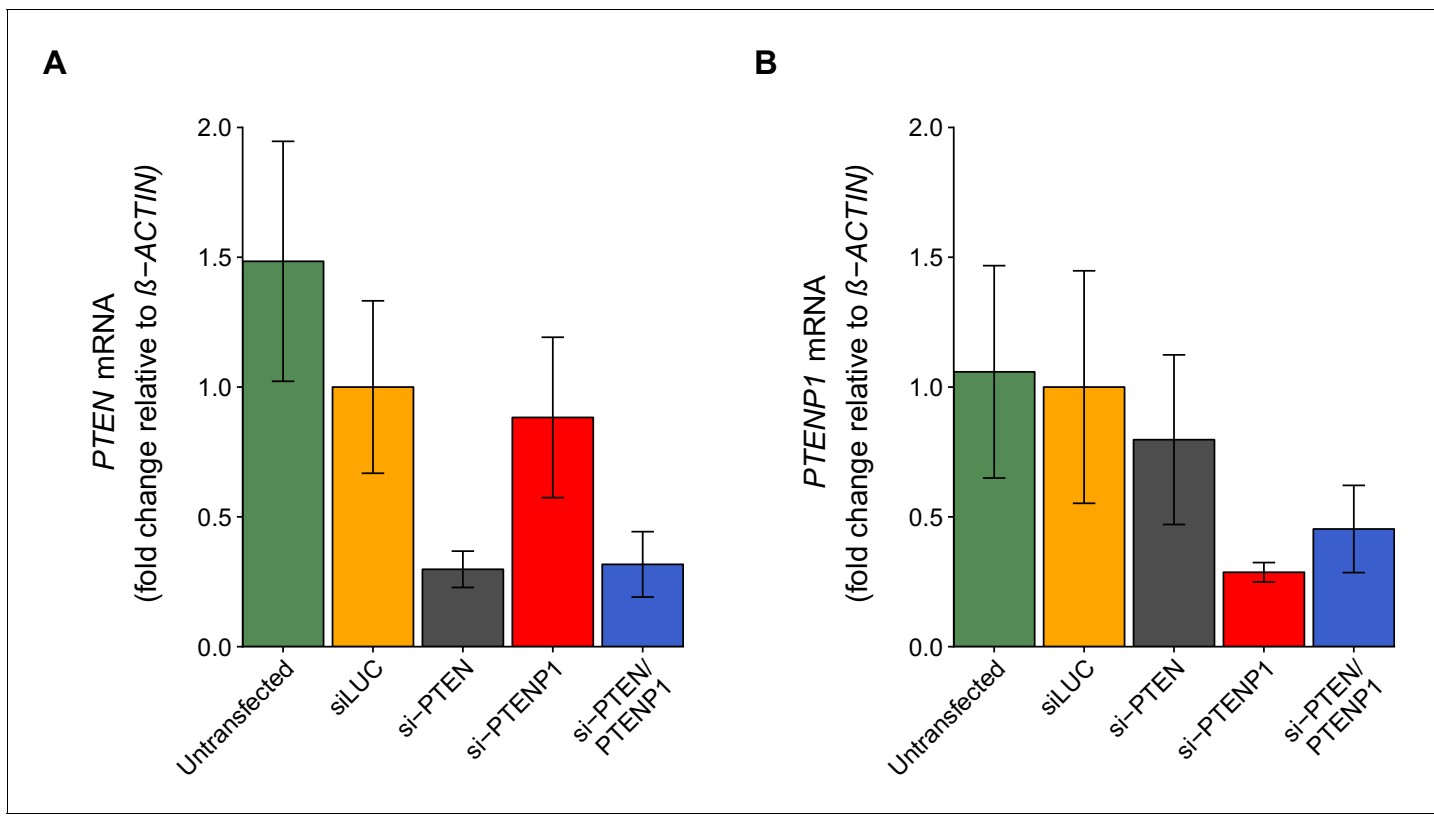

**Figure 2.** *PTEN* and *PTENP1* abundance in DU145 cells depleted of *PTEN* and/or *PTENP1*. DU145 cells were transfected with either a non-targeting siRNA (siLUC), si-PTEN, si-PTENP1, or an siRNA pool targeting *PTEN* and *PTENP1* (si-PTEN/PTENP1), or not transfected. Total RNA was isolated 24 hr later and qRT-PCR analysis was performed to detect *PTEN*, *PTENP1*, and *ß-ACTIN* levels. (A) Fold change in *PTEN* expression (*PTEN/ß-ACTIN*) is presented for each condition relative to siLUC cells. Means reported and error bars represent *SD* from five independent biological repeats. Two-way ANOVA interaction between *PTEN* (targeted or not-targeted) and *PTENP1* (targeted or not-targeted) on *PTEN* expression: $F_{(1,16)} = 0.41$, $p=0.532$; main effect of *PTEN*: $F_{(1,16)} = 35.5$, $p=2.01 \times 10^{-5}$; main effect of *PTENP1*: $F_{(1,16)} = 0.21$, $p=0.651$. Planned contrasts between siLUC and si-PTEN for *PTEN* expression: $t(16) = 4.66$, uncorrected $p=0.00026$ with *a priori* Bonferroni adjusted significance threshold of 0.0083, Bonferroni corrected $p=0.0016$; siLUC and si-PTENP1: $t(16) = 0.78$, uncorrected $p=0.448$, Bonferroni corrected $p>0.99$; siLUC and si-PTEN/PTENP1: $t(16) = 4.54$, uncorrected $p=0.00034$, Bonferroni corrected $p=0.0020$. (B) Fold change in *PTENP1* expression (*PTENP1/ß-ACTIN*) is presented for each condition relative to siLUC cells. Means reported and error bars represent *SD* from five independent biological repeats. Two-way ANOVA interaction on *PTENP1* expression: $F_{(1,16)} = 2.03$, $p=0.174$; main effect of *PTEN*: $F_{(1,16)} = 0.019$, $p=0.891$; main effect of *PTENP1*: $F_{(1,16)} = 16.6$, $p=0.00088$. Planned contrasts between siLUC and si-PTEN for *PTENP1* expression: $t(16) = 1.10$, uncorrected $p=0.286$ with *a priori* Bonferroni adjusted significance threshold of 0.0083, Bonferroni corrected $p>0.99$; siLUC and si-PTENP1: $t(16) = 3.89$, uncorrected $p=0.0013$, Bonferroni corrected $p=0.0079$; siLUC and si-PTEN/PTENP1: $t(16) = 2.98$, uncorrected $p=0.0089$, Bonferroni corrected $p=0.053$. Additional details for this experiment can be found at https://osf.io/4uard/.

The online version of this article includes the following figure supplement(s) for figure 2:

**Figure supplement 1.** *PTEN* and *PTENP1* abundance using *36B4* to normalize expression.

replication attempt is a possible reason for the differences in outcomes between the two studies. The experiment that assessed the level of knockdown in this replication is described in the section below (*Figure 2*) and compared to the results reported in the original study. Various levels of down-regulation can give various phenotype effects, thus a higher level of knockdown might be required to observe an effect with this experimental design. Although, observing different outcomes are informative to establish the range of conditions under which a given phenotype can be observed (*Bailoo et al., 2014*).

The analysis plan specified in the Registered Report (*Khan et al., 2015*) proposed to compare the Avg Day 5# for the cells transfected with the various siRNAs. To test if downregulation of *PTEN* and/or *PTENP1*, was effective in increasing DU145 cell proliferation we performed an analysis of variance (ANOVA). The ANOVA result was statistically significant for the *PTENP1* main effect ($F_{(1,16)}$ = 13.8, $p$=0.0019). Thus, the null hypothesis that there is no difference in cell proliferation when *PTENP1* is downregulated, whether there was or was not downregulation of *PTEN*, can be rejected. The main effect for *PTEN* downregulation ($F_{(1,16)}$ = 0.15, $p$=0.703) and the interaction ($F_{(1,16)}$ = 0.02, $p$=0.878) were not statistically significant. We planned to conduct five comparisons using the Bonferroni correction to adjust for multiple comparisons, making the *a priori* adjusted significance threshold 0.01. According to this criterion, none of the planned comparisons were statistically significant (see *Figure 1* figure legend). The same analysis was conducted using area under the curve (AUC) during the timecourse for each biological repeat, which gave similar results (*Figure 1—figure supplement 1B*). To summarize, for this experiment we found results that were generally in the same direction as the original study (i.e., increased cell proliferation when *PTEN* and/or *PTENP1* were downregulated compared to non-targeting siRNA) and not statistically significant where predicted.

## Quantitative PCR following transfection with siRNA against PTEN and/or PTENP1

To determine the knockdown efficiency of *PTEN* and *PTENP1* in DU145 cells we performed qRT-PCR following transfection with the same siRNAs targeting *PTEN* and/or *PTENP1* described above. This experiment also tested whether *PTENP1* knockdown resulted in decreased *PTEN* mRNA, and vice versa. This is similar to what was reported in Figure 2G of *Poliseno et al. (2010)* and described in Protocol 3 in the Registered Report (*Khan et al., 2015*). We found that compared to cells transfected with non-targeting siRNA (mean relative expression = 1.00), cells transfected with *PTEN* specific siRNA or the siRNA pool targeting *PTEN* and *PTENP1* resulted in decreased *PTEN* expression with relative mean values of 0.298 and 0.317, respectively, which were both statistically significant (non-targeting vs. si-PTEN: $t_{(16)}$ = 4.66, uncorrected $p$=0.00026, corrected $p$=0.0016; non-targeting vs. si-PTEN/PTENP1: $t_{(16)}$ = 4.54, uncorrected $p$=0.00034, corrected $p$=0.0020) (*Figure 2A*). Cells transfected with *PTENP1* specific siRNA, however, resulted in a slight decrease in *PTEN* expression (mean relative expression = 0.883), which was not statistically significant ($t_{(16)}$ = 0.78, uncorrected $p$=0.448, corrected $p$>0.99). Similarly, cells transfected with *PTENP1* specific siRNA or the siRNA pool targeting *PTEN* and *PTENP1*, resulted in decreased relative *PTENP1* expression with relative mean values of 0.287 and 0.453, respectively; while *PTENP1* expression was slightly decreased following transfection with *PTEN* specific siRNA (mean relative expression = 0.797) (*Figure 2B*). According to the criterion predefined in the Registered Report, cells transfected with *PTENP1* specific siRNA were statistically significant (non-targeting vs. si-PTENP1: $t_{(16)}$ = 3.89, uncorrected $p$=0.0013, corrected $p$=0.0079), while the other two comparisons were not (non-targeting vs. si-PTEN/PTENP1: $t_{(16)}$ = 2.98, uncorrected $p$=0.0089, corrected $p$=0.053; non-targeting vs. si-PTEN: $t_{(16)}$ = 1.10, uncorrected $p$=0.286, corrected $p$>0.99). The original study reported statistically significant decreases in *PTEN* and *PTENP1* expression for all of the comparisons between cells transfected with non-targeting siRNA and either si-PTEN, si-PTENP1, or the siRNA pool targeting *PTEN* and *PTENP1*. In the original study *PTEN* expression was decreased with relative mean values of ~0.12, ~0.38, and ~0.19 and *PTENP1* expression was decreased with relative mean values of ~0.54, ~0.27, and ~0.48 for si-PTEN, si-PTENP1, and si-PTEN/PTENP1, respectively (*Poliseno et al., 2010*). Further, these replication and original results used *ß-ACTIN* as the internal standard to normalize *PTEN* and *PTENP1* expression. We also normalized the *PTEN* and *PTENP1* levels from this experiment to *36B4*, as suggested during peer review of the Registered Report (*Davis, 2015*), and found similar results (*Figure 2—figure supplement 1*). To summarize, for this

experiment we found results that were in the same direction as the original study and statistically significant for confirming knockdown of *PTEN* or *PTENP1* with target specific siRNA; however results were not statistically significant for other comparisons (i.e., *PTEN* mRNA levels following *PTENP1* knockdown, and vice versa).

There are a number of factors that could affect these experiments and should be considered when interpreting these data. Although qRT-PCR is a common laboratory technique there are a number of steps and reagents that can vary in experimental protocols and data analyses among different laboratories (*Kuang et al., 2018*; *Miranda and Steward, 2017*; *Taylor et al., 2019*). While this replication attempted to mirror the approach taken in the original study it was limited to what was obtainable from the original paper and through communication with the original authors. Similarly, the raw data (e.g., levels of *ß-ACTIN* in the PCR reaction) from the original study were unknown and thus can not be directly compared to this replication attempt. Additionally, cell to cell contact has been reported to active microRNA biogenesis globally (*Hwang et al., 2009*). Thus, the level of cellular confluence, which was not able to be compared between the original study and this replication attempt, is another factor that needs to be considered. Finally, the higher variability observed in this study is one of the factors that could influence if statistical significance is reached, particularly since the sample size of this replication attempt was determined *a priori* to detect the effect based on the originally reported data.

## Western blot of cells transfected with siRNA

We also examined the knockdown efficiency and impact of *PTENP1* knockdown on PTEN protein levels using the same siRNAs targeting *PTEN* and/or *PTENP1* described above and the same PTEN antibody as the original study. This is similar to what was reported in Figure 2H of *Poliseno et al. (2010)* and described in Protocol 4 in the Registered Report (*Khan et al., 2015*). We found that compared to cells transfected with non-targeting siRNA (mean relative expression = 1.00), cells transfected with *PTEN* specific siRNA, *PTENP1* specific siRNA, or the siRNA pool targeting *PTEN* and *PTENP1* resulted in decreased PTEN expression with relative mean values of 0.41, 0.65, and 0.17, respectively (*Figure 3*). The original study reported that compared to non-targeting siRNA (relative expression = 1.00) the relative PTEN expression was 0.50 for cells transfected with *PTEN* specific siRNA, 0.60 for cells transfected with *PTENP1* specific siRNA, and 0.10 for cells transfected with the siRNA pool targeting *PTEN* and *PTENP1*.

To compare the relative PTEN expression among the various conditions, we planned to conduct five comparisons using the Bonferroni correction to adjust for multiple comparisons, making the *a priori* adjusted significance threshold 0.01. Using the Wilcoxon-Mann-Whitney test we found statistically significant differences between non-targeting siRNA and si-PTEN ($U = 25$, uncorrected $p=0.0079$, corrected $p=0.040$), non-targeting siRNA and si-PTEN/PTENP1 ($U = 25$, uncorrected $p=0.0079$, corrected $p=0.040$), and si-PTENP1 and si-PTEN/PTENP1 ($U = 0$, uncorrected $p=0.0079$, corrected $p=0.040$). The comparisons between non-targeting siRNA and si-PTENP1 ($U = 21$, uncorrected $p=0.095$, corrected $p=0.476$) and si-PTEN and si-PTEN/PTENP1 ($U = 6$, uncorrected $p=0.222$, corrected $p>0.99$) were not statistically significant. To summarize, for this experiment we found results that were in the same direction as the original study with statistically significant decreases in relative PTEN expression from cells transfected with si-PTEN or si-PTEN/PTENP1 compared to non-targeting siRNA or si-PTENP1, while the differences in relative PTEN expression between non-targeting siRNA and si-PTENP1, and si-PTEN and si-PTEN/PTENP1, were not statistically significant.

## Quantitative PCR following *PTEN* 3′ UTR transfection

DU145 cells were transfected with a plasmid designed to express the *PTEN* 3′UTR to test if the *PTEN* 3′UTR exerted a biological role on *PTENP1*. This was described in Protocol 5 in the Registered Report (*Khan et al., 2015*) and is similar to Figure 4A of *Poliseno et al. (2010)* that reported *PTEN* 3′UTR derepressed *PTENP1* abundance. We found *PTENP1* abundance was similar and not statistically significant (two sample *t* test: $t(4) = 0.46$, $p=0.671$) between cells transfected with a vector control plasmid (mean relative expression = 1.00) and cells transfected with *PTEN* 3′UTR (mean relative expression = 0.95) (*Figure 4*). The original study reported a statistically significant increase in *PTENP1* expression when cells were transfected with a plasmid that expressed the *PTEN* 3′UTR (mean relative expression = ~3.88) compared to vector control (mean relative expression = 1.00)

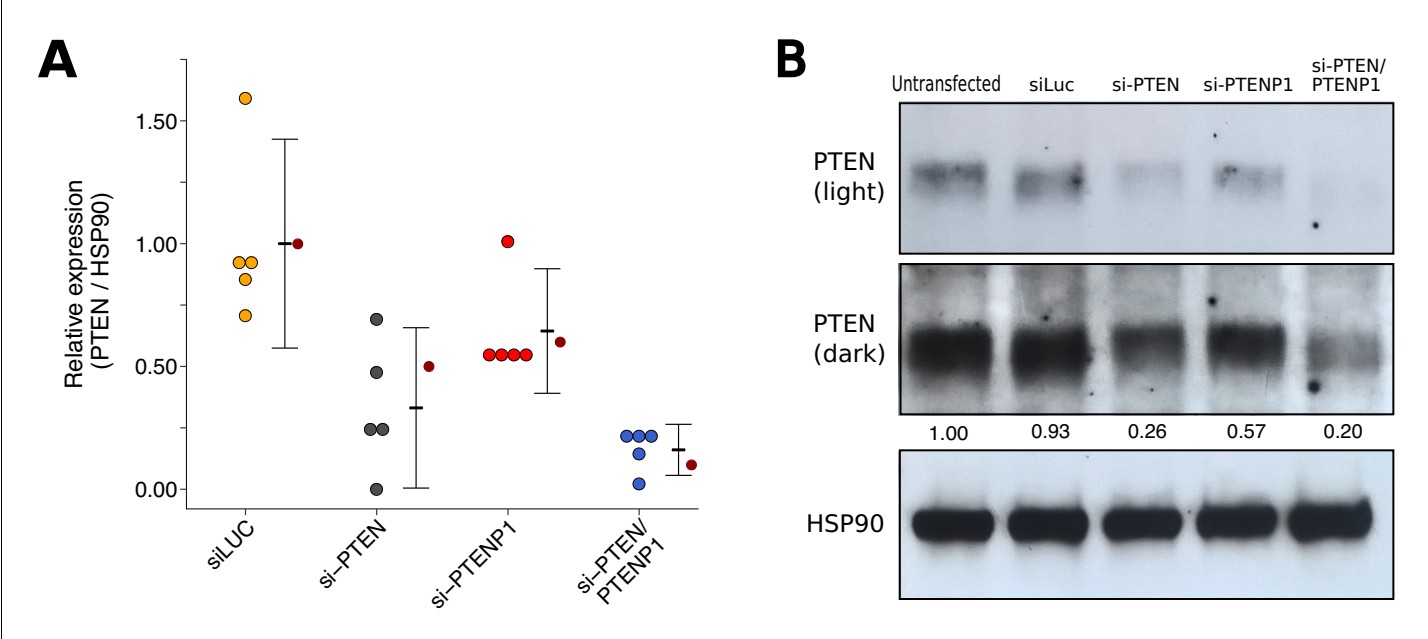

**Figure 3.** PTEN expression in DU145 cells depleted of *PTEN* and/or *PTENP1*. DU145 cells were transfected with either a non-targeting siRNA (siLUC), si-PTEN, si-PTENP1, or an siRNA pool targeting *PTEN* and *PTENP1* (si-PTEN/PTENP1), or not transfected. Cells were harvested 48 hr later for Western blot analysis. (**A**) Relative protein expression (PTEN/HSP90) are presented for each condition. Western blot bands were quantified, PTEN levels were normalized to HSP90, then for each biological repeat values were normalized to the untransfected condition with protein expression presented relative to siLuc. Dot plot of independent biological repeats (n = 5), means reported as crossbars and error bars represent 95% CI. Data reported in Figure 2H of *Poliseno et al. (2010)* is displayed as a single point (small dark red circle) for comparison. Planned comparisons (two-tailed Wilcoxon-Mann-Whitney tests): siLUC and si-PTEN: $U = 25$, uncorrected $p=0.0079$ with *a priori* Bonferroni adjusted significance threshold of 0.01, Bonferroni corrected $p=0.040$; siLUC and si-PTENP1: $U = 21$, uncorrected $p=0.095$, Bonferroni corrected $p=0.476$; siLUC and si-PTEN/PTENP1: $U = 25$, uncorrected $p=0.0079$, Bonferroni corrected $p=0.040$; si-PTEN/PTENP1 and si-PTEN: $U = 6$, uncorrected $p=0.222$, Bonferroni corrected $p>0.99$; si-PTEN/PTENP1 and si-PTENP1: $U = 0$, uncorrected $p=0.0079$, Bonferroni corrected $p=0.040$. (**B**) Representative Western blots probed with an anti-PTEN antibody (two exposures presented to facilitate detection) and anti-HSP90 antibody. Relative PTEN/HSP90 expressions are reported below PTEN images. Additional details for this experiment can be found at https://osf.io/re87y/.

(*Poliseno et al., 2010*). Similar to the above experiments, these results used *ß-ACTIN* as the internal standard to normalize *PTENP1* expression; however when we used *36B4* to normalize *PTENP1* levels we found similar results (*Figure 4—figure supplement 1*). To summarize, for this experiment we found results that were not in the same direction as the original study and not statistically significant where predicted. The level of *PTEN* 3'UTR expression was not determined for this experiment and thus is a possible reason for differences in outcomes between the original study and this replication attempt. Additionally, as noted above, potential differences between the original study and this replication attempt are factors that need to be considered when interpreting these results. These include, but are not limited to, variations in expression levels of the internal standards used to normalize *PTENP1* expression, differences in observed biological variability between the original study and this replication attempt, and cellular confluence, which has been reported to modify the levels of microRNA expression and thus their target constructs (*Hwang et al., 2009*).

## Cell growth assay following *PTEN* 3′ UTR transfection

The impact of *PTEN* 3'UTR overexpression on DU145 cell proliferation was also examined. This is similar to what was reported in Figure 4A of *Poliseno et al. (2010)* and described in Protocol 6 in the Registered Report (*Khan et al., 2015*). Using the crystal violet assay, proliferative activity was determined each day after transfection for a total of 5 days with results presented relative to the values at the start of the timecourse for each condition (same approach as described above) (*Figure 5*, *Figure 5—figure supplement 1A*). We found that compared to cells transfected with a vector control plasmid (Avg Day 5# = 3.79), there was growth inhibition observed when cells were transfected with *PTEN* 3'UTR (Avg Day 5# = 2.72), which was statistically significant (two sample *t* test: *t*(4) =

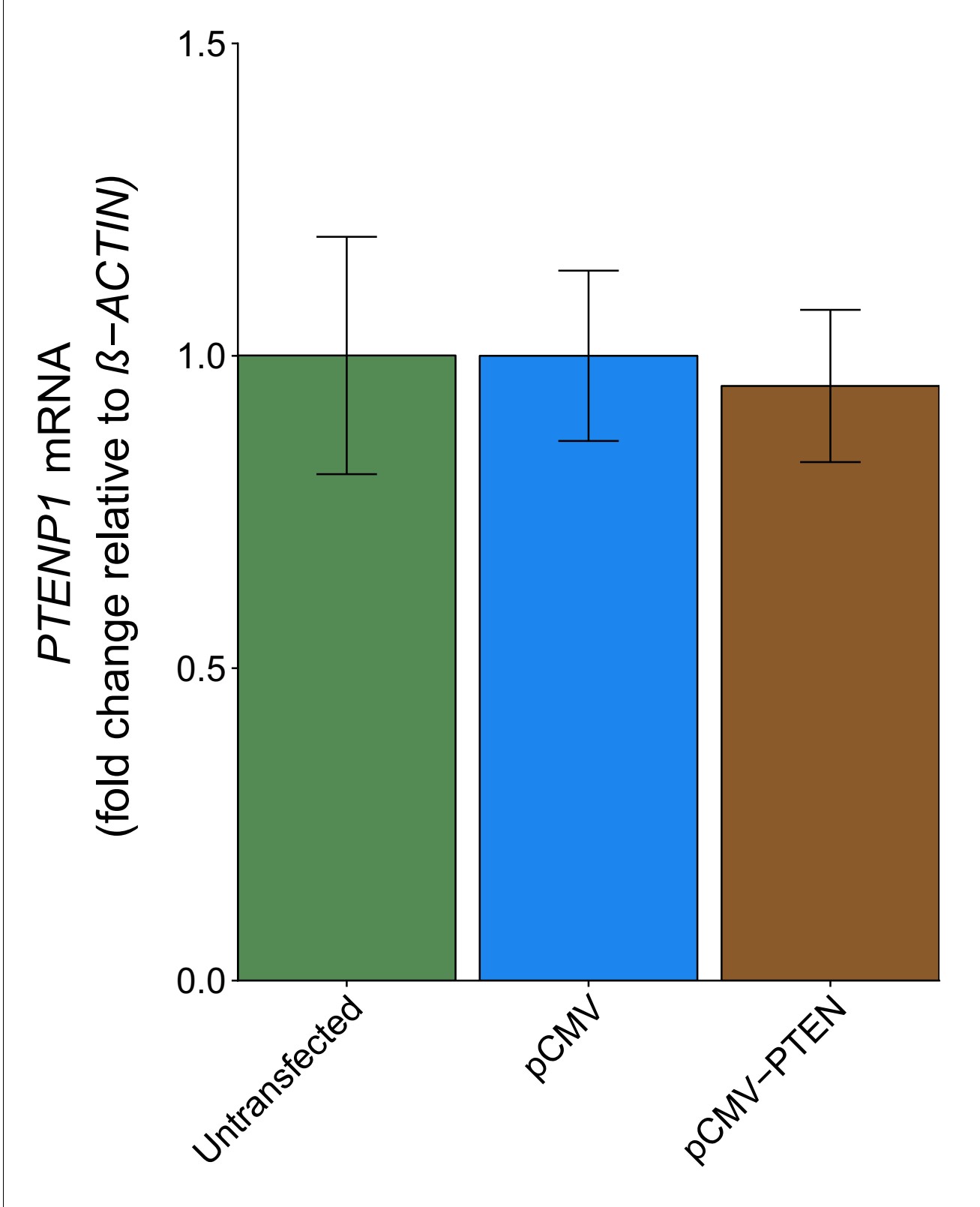

**Figure 4.** *PTENP1* abundance in DU145 cells expressing *PTEN* 3'UTR. DU145 cells were transfected with either a vector control plasmid (pCMV) or a plasmid to express *PTEN* 3'UTR (pCMV-*PTEN*), or not transfected. Total RNA was isolated 24 hr later and qRT-PCR analysis was performed to detect *PTENP1* and *ß-ACTIN* levels. Fold change in *PTENP1* expression (*PTENP1/ß-ACTIN*) is presented for each condition relative to pCMV
*Figure 4 continued on next page*

*Figure 4 continued*

transfected cells. Means reported and error bars represent *SD* from three independent biological repeats. Unpaired two-tailed Student's *t* test between pCMV and pCMV-*PTEN*: *t*(4) = 0.46, *p*=0.671. Additional details for this experiment can be found at https://osf.io/rkuxh/.

The online version of this article includes the following figure supplement(s) for figure 4:

**Figure supplement 1.** *PTENP1* abundance using *36B4* to normalize expression.

9.25, *p*=0.00076). The same analysis was conducted using AUC during the 5 day timecourse for each biological repeat, which gave similar results (*Figure 5—figure supplement 1B*). In addition, untransfected cells displayed a cell proliferation profile similar to the vector control (*Figure 5*, *Figure 5— figure supplement 1*). In the original study, overexpression of *PTEN* 3'UTR inhibited cell proliferation (Avg Day 5# = ~5.24) compared to vector control (Avg Day 5# = ~7.17) (*Poliseno et al., 2010*). Interestingly, the growth profiles of the control conditions differed between this experiment and the

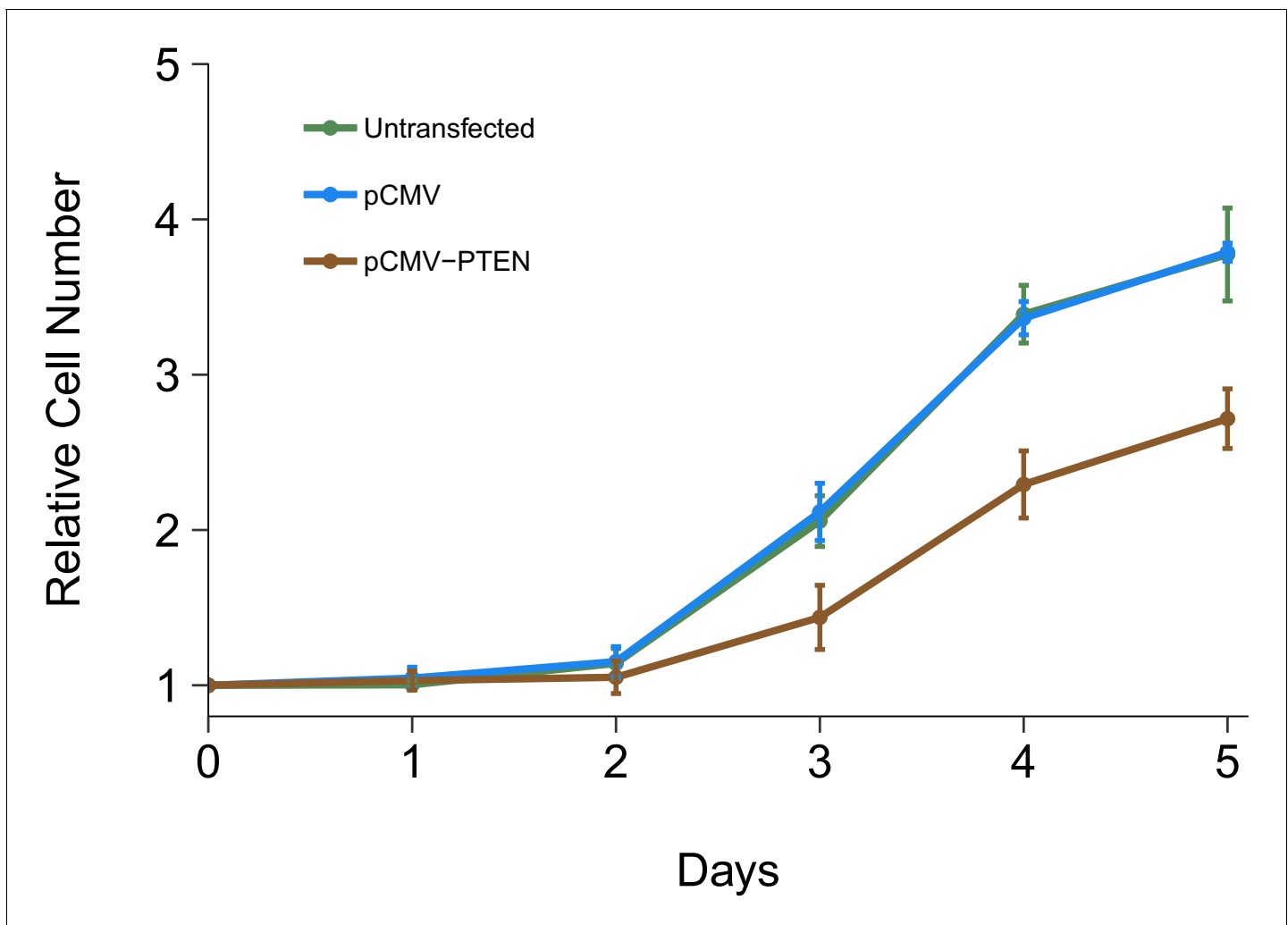

**Figure 5.** Cell growth of DU145 cells expressing *PTEN* 3'UTR. DU145 cells were transfected with either vector control plasmid (pCMV) or a plasmid to express PTEN 3'UTR (pCMV-*PTEN*), or not transfected. Crystal violet proliferation assays were performed each day as indicated starting the day after transfection. Relative cell number was calculated relative to the average Day 0 values for each condition. Means reported and error bars represent *SD* from three independent biological repeats. Unpaired two-tailed Student's *t* test between pCMV and pCMV-*PTEN* on Day five relative cell numbers: *t*(4) = 9.25, *p*=0.00076. Additional details for this experiment can be found at https://osf.io/jgp6n/.

The online version of this article includes the following figure supplement(s) for figure 5:

**Figure supplement 1.** Alternative visualization of cell growth.

experiment described above where DU145 cells were transfected with different siRNAs. Importantly, in this replication attempt this experiment and the cell growth assay presented in *Figure 1* were conducted by two independent labs, therefore operator and/or environmental differences could be contributing factors. This could also be the result of cells being in the lag phase of cell growth at the start of the experiment instead of exponential phase growth, which could be caused by the seeding density and/or the *in vitro* age of the cultures (*Selli et al., 2016*). The original study also reported different cell growth profiles between the control conditions in the knockdown and overexpression experiments. Furthermore, the growth inhibition observed in the original study in cells that over-expressed *PTEN* 3'UTR was correlated with an increase in *PTENP1* abundance, which, as described above (*Figure 4*), was not observed in this replication attempt. To summarize, for this experiment we found results that were in the same direction as the original study and statistically significant where predicted.

## Meta-analyses of original and replication effects

We performed a meta-analysis using a random-effects model, where possible, to combine each of the effects described above as pre-specified in the confirmatory analysis plan (*Khan et al., 2015*). To provide a standardized measure of the effect, a common effect size was calculated for each effect from the original and replication studies. Cohen's *d* is the standardized difference between two means using the pooled sample standard deviation, while the effect size Glass' delta is standardized difference between two means using the standard deviation of only the control group. Glass' delta was used when the variance between the two conditions were not equal, which occurred in the original study for some of the experiments. The estimate of the effect size of one study, as well as the associated uncertainty (i.e., confidence interval), compared to the effect size of the other study provides one approach to compare the original and replication results (*Errington et al., 2014*; *Valentine et al., 2011*). Importantly, the width of the confidence interval (CI) for each study is a reflection of not only the confidence level (e.g., 95%), but also variability of the sample (e.g., *SD*) and sample size.

There were five comparisons of the Day five relative cell number data from DU145 cells that were transfected with siLUC, si-PTEN, si-PTENP1, or si-PTEN/PTENP1, which were reported in *Figure 1* of this study and Figure 2F of *Poliseno et al. (2010)*. In all comparisons the results were consistent when considering the direction of the effect; however the effect size point estimate of each study was not within the CI of the other study (*Figure 6A*). Further, the meta-analyses were not statistically significant for any of the comparisons (see *Figure 6A* figure legend). For three of the comparisons, the large CI of the meta-analyses along with statistically significant Cochran's *Q* tests (siLUC and si-PTEN, $p=0.0253$; siLUC and si-PTEN/PTENP1, $p=0.0349$; si-PTEN/PTENP1 and si-PTENP1, $p=0.0235$) suggest heterogeneity between the original and replication studies.

*PTEN* and *PTENP1* mRNA expression were also examined in DU145 cells transfected with siLUC, si-PTEN, si-PTENP1, or si-PTEN/PTENP1, reported in *Figure 2* of this study and Figure 2G of *Poliseno et al. (2010)*, in which a total of six comparisons were made. With the *PTEN* mRNA expression data, two comparisons (siLUC and si-PTEN; siLUC and si-PTEN/PTENP1) had effects in the same direction for both the original study and this replication attempt, with the CI of each study encompassing the effect size point estimate of the other study (*Figure 6B*). The third comparison (siLUC and si-PTENP1) had effects in the same direction for both studies, but the point estimate of the original study was not within the CI of the replication. Furthermore, the meta-analyses for the first two comparisons were statistically significant (siLUC and si-PTEN, $p=0.0013$; siLUC and si-PTEN/PTENP1, $p=0.0019$), while the meta-analysis of the third comparison was not statistically significant (siLUC and si-PTENP1, $p=0.244$). These meta-analyses suggest transfection with si-PTEN or si-PTEN/PTENP1 decreased *PTEN* expression, while with the si-PTENP1 transfection the null hypothesis that there is no difference in *PTEN* mRNA expression can not be rejected. With the *PTENP1* mRNA expression data, each of the three comparisons were consistent with respect to the direction of the effect; however, for each comparison the point estimate of the replication fell within the CI of the original study, but not vice versa. For the meta-analyses, two comparisons were not found to be statistically significant (siLUC and si-PTEN, $p=0.225$; siLUC and si-PTENP1, $p=0.060$), while one was (siLUC and si-PTEN/PTENP1, $p=0.049$). These meta-analyses suggest transfection with si-PTEN/PTENP1 decreases *PTENP1* expression, while with the two null hypotheses that there are no differences in *PTENP1* expression after transfection with si-PTEN or si-PTENP1 can not be rejected.

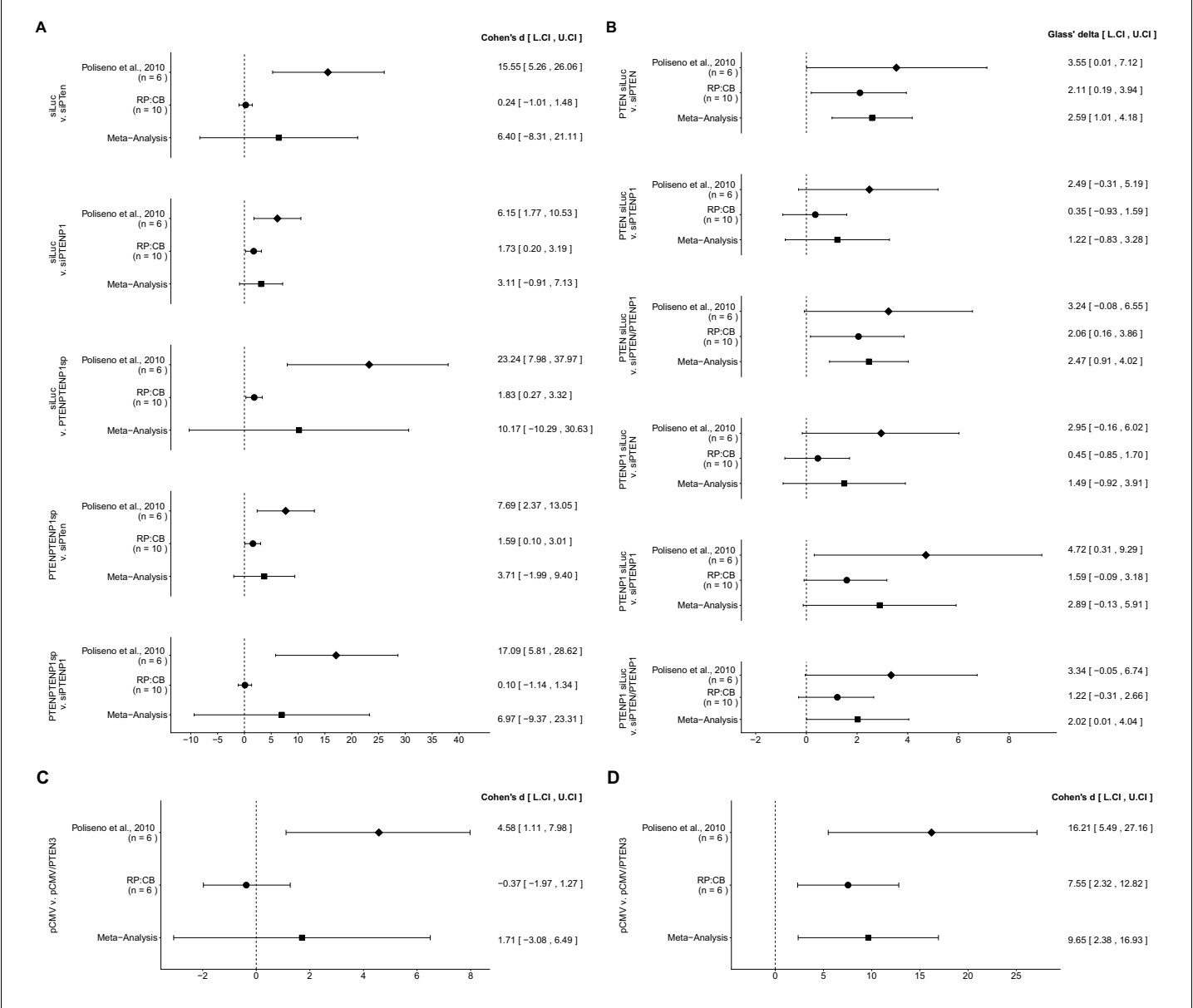

**Figure 6.** Meta-analyses of each effect. Effect size and 95% confidence interval are presented for *Poliseno et al. (2010)*, this replication study (RP:CB), and a random effects meta-analysis of those two effects. Cohen's *d* and Glass' delta are standardized differences between the two indicated measurements with the calculated effects for the original study effects reported as positive values. Sample sizes used in *Poliseno et al. (2010)* and RP: CB are reported under the study name. (**A**) These effects are related to the change in Day 5 relative cell numbers between the conditions reported in *Figure 1* of this study and Figure 2F of *Poliseno et al. (2010)*. Meta-analysis *p* values: siLUC and si-PTEN (*p*=0.394); siLUC and si-PTENP1 (*p*=0.129); siLUC and si-PTEN/PTENP1 (*p*=0.330); si-PTEN/PTENP1 and si-PTEN (*p*=0.202); si-PTEN/PTENP1 and si-PTENP1 (*p*=0.403). (**B**) These effects are related to the fold change differences in *PTEN* and *PTENP1* expression between the conditions reported in *Figure 2* of this study and Figure 2G of *Poliseno et al. (2010)*. Meta-analysis *p* values: *PTEN* expression between siLUC and si-PTEN (*p*=0.0013); *PTEN* expression between siLUC and si-PTENP1 (*p*=0.244); *PTEN* expression between siLUC and si-PTEN/PTENP1 (*p*=0.0019); *PTENP1* expression between siLUC and si-PTEN (*p*=0.225); *PTENP1* expression between siLUC and si-PTENP1 (*p*=0.060); *PTENP1* expression between siLUC and si-PTEN/PTENP1 (*p*=0.049). (**C**) These effects are related to the fold change differences in *PTENP1* expression between pCMV and pCMV-*PTEN*-3'UTR reported in *Figure 4* of this study and Figure 4A of *Poliseno et al. (2010)* (meta-analysis *p*=0.485). (**D**) These effects are related to the change in Day 5 relative cell numbers between pCMV and pCMV-*PTEN*-3'UTR reported in *Figure 5* of this study and Figure 4A of *Poliseno et al. (2010)* (meta-analysis *p*=0.0093). Additional details for these meta-analyses can be found at https://osf.io/9yh6p/.

The original study results of the Western blot data examining PTEN protein levels after transfection with siRNAs targeting *PTEN* and/or *PTENP1* were reported for a single representative image (*Figure 2H*; *Poliseno et al., 2010*). These values were compared to the values reported in this replication attempt and for each of the conditions tested the values reported in the original study were within the CI of this replication attempt (*Figure 3A*).

The effect sizes calculated from the *PTENP1* mRNA expression data between DU145 cells transfected with pCMV or pCMV-*PTEN*-3'UTR, which were reported in *Figure 4* of this study and Figure 4A of *Poliseno et al. (2010)*, were in opposite directions between the two studies and the effect size point estimate of each study was not within the CI of the other study (*Figure 6C*). Furthermore, the meta-analysis was not statistically significant ($p$=0.485) with a large CI and a statistically significant Cochran's $Q$ test ($p$=0.0425) that suggests heterogeneity between the original and replication studies.

For the Day five relative cell number data from DU145 cells transfected with pCMV or pCMV-*PTEN*-3'UTR, reported in *Figure 5* of this study and Figure 4A of *Poliseno et al. (2010)*, the calculated effect sizes were consistent when considering direction (*Figure 6D*). The replication effect size point estimate fell within the CI of the original study, but not vice versa. A meta-analysis of these effects were found to be statistically significant ($p$=0.0093), suggesting transfection of DU145 cells with a plasmid designed to express the *PTEN* 3'UTR inhibited growth compared to a vector control plasmid.

This direct replication provides an opportunity to understand the present evidence of these effects. Any known differences, including reagents and protocol differences, were identified prior to conducting the experimental work and described in the Registered Report (*Khan et al., 2015*). However, this is limited to what was obtainable from the original paper and through communication with the original authors, which means there might be particular features of the original experimental protocol that could be critical, but unidentified. So while some aspects, such as cell lines, number of cells plated, and the specific siRNA sequences were maintained, others were unknown or not easily controlled for. These include variables such as cell line genetic drift (*Hughes et al., 2007*; *Kleensang et al., 2016*), and impacts of atmospheric oxygen on cell viability and growth (*Boregowda et al., 2012*). Whether these or other factors influence the outcomes of this study is open to hypothesizing and further investigation, which is facilitated by direct replications and transparent reporting.

## Materials and methods

**Key resources table**

| Reagent type (species) or resource | Designation | Source or reference | Identifiers | Additional information |
|---|---|---|---|---|
| Cell line (*H. sapiens*, male) | DU145 | ATCC | cat#:HTB-81; RRID:CVCL_0105 | |
| Sequence-based reagent | siGlo RISC-free siRNA | Dharmacon | cat#:D-001600–01 | |
| Sequence-based reagent | siGENOME non-targeting siRNA #2 | Dharmacon | cat#:D-001210–02 | |
| Sequence-based reagent | *PTEN*-specific SMARTpool | Dharmacon | | Reported in Supplementary Figure 6 (DOI: 10.1038/nature09144) |
| Sequence-based reagent | *PTENP1*-specific SMARTpool | Dharmacon | | Reported in Supplementary Figure 6 (DOI: 10.1038/nature09144) |
| Sequence-based reagent | ON-TARGETplus siPTEN SMARTpool | Dharmacon | cat#:L-003023–00 | |
| Recombinant DNA reagent | pCMV | Agilent Technologies | cat#:240071 | |
| Recombinant DNA reagent | pCMV-*PTEN*-3'UTR | This paper | RRID:Addgene_97204 | |

*Continued on next page*

*Continued*

| Reagent type (species) or resource | Designation | Source or reference | Identifiers | Additional information |
|---|---|---|---|---|
| Antibody | rabbit anti-PTEN | Cell Signaling Technology | cat#:9559; clone:138G5; RRID:AB_390810 | 1:1000 dilution |
| Antibody | mouse anti-HSP90 | BD Biosciences | cat#:610419; clone:68; RRID:AB_397798 | 1:1000 dilution |
| Antibody | HRP-conjugated goat anti-rabbit | Cell Signaling Technology | cat#:7074; RRID:AB_2099233 | 1:20,000 dilution |
| Antibody | HRP-conjugated horse anti-mouse | Cell Signaling Technology | cat#:7076; RRID:AB_330924 | 1:20,000 dilution |
| Software, algorithm | CFX Manager | BioRad | RRID:SCR_017251 | version 3.1.1517.0823 |
| Software, algorithm | ImageJ | DOI: 10.1038/nmeth.2089 | RRID:SCR_003070 | version 1.50a |
| Software, algorithm | SoftMax Pro data acquisition and analysis | Molecular Devices | RRID:SCR_014240 | version 4.6 |
| Software, algorithm | R Project for statistical computing | https://www.r-project.org | RRID:SCR_001905 | version 3.5.1 |

As described in the Registered Report (*Khan et al., 2015*), we attempted a replication of the experiments reported in Figures 2F–H and 4A of *Poliseno et al. (2010)*. A detailed description of all protocols can be found in the Registered Report (*Khan et al., 2015*) and are described below with additional information not listed in the Registered Report, but needed during experimentation.

## Cell culture

DU145 cells (ATCC, cat# HTB-81, RRID:CVCL_0105) were grown in RPMI 1640 supplemented with 10% Fetal Bovine Serum (FBS), 100 U/ml penicillin, 100 μg/ml streptomycin, and 2 mM L-glutamine. Cells were maintained at 37°C in a humidified atmosphere at 6% $CO_2$. Quality control data for the experiments presented in *Figures 1–3* are available at https://osf.io/zxuya/, while data for the experiments presented in *Figures 4*, *5* are available at https://osf.io/b4rt2/. This includes results confirming the cell lines were free of mycoplasma contamination as well as STR DNA profiling of the cell lines, which were confirmed to be the indicated cell lines when queried against STR profile databases. Test for mycoplasma was performed by DDC Medical (Fairfield, Ohio) or using the MycoProbe Mycoplasma Detection Assay Kit (R and D Systems, cat# CUL0001B, lot# P104920). STR profile was performed by DDC Medical or Genetica Cell Line Testing (Burlington, North Carolina).

## siRNA transfection and crystal violet proliferation assay

DU145 cells were seeded at $1.5 \times 10^5$ cells per well in a 12 well plate and grown overnight. The next day cells were transfected using Dharmafect according to manufacturer's instructions with 25 nM siGlo RISC-free siRNA (Dharmacon, cat# D-001600–01), 100 nM siGENOME non-targeting siRNA #2 (siLuc; Dharmacon, cat# D-001210–02), 100 nM *PTEN*-specific SMARTpool siPTEN; Dharmacon; sequences reported in Supplementary Figure 6 of *Poliseno et al. (2010)*, 100 nM *PTENP1*-specific SMARTpool siPTENP1; Dharmacon; sequences reported in Supplementary Figure 6 of *Poliseno et al. (2010)*, 100 nM ON-TARGETplus siPTEN SMARTpool (siPTEN/PTENP1; Dharmacon, cat# L-003023–00), or left untransfected. After 24 hr, microscopy was performed on cells transfected with siGlo and untransfected cells to ascertain transfection efficiency, which was observed to be >90%. Cells transfected with siLuc, siPTEN, siPTENP1, siPTEN/PTENP1, or untransfected were plated at 8,000 cells per well (in triplicate) in 12 well plates (enough for 6 days of measurements) with 2 ml of growth medium. Starting the day after plating (designated day 0), every 24 hr the crystal violet assay was performed as described in the Registered Report. Additionally, two additional cohorts (wells in triplicate) were included: unseeded (i.e., no cells) wells were treated with the Crystal

Violet reagent (Blank treated) and unseeded wells left entirely untreated (Blank untreated). The average value from Blank treated was subtracted from each data point to correct for background staining of crystal violet to the plastic dish. For each independent biological repeat, average absorbance ($OD_{590}$) for each condition was normalized by dividing the average absorbance of each day to the average absorbance for day 0 to calculate relative cell number. Area under the curve (AUC) was calculated for each condition of each biological repeat. Data files are available at https://osf.io/xyefm/.

### siRNA transfection and quantitative PCR

DU145 cells were seeded and transfected as described above in 'siRNA transfection and crystal violet proliferation assay'. 24 hr after transfection, total RNA was extracted with TRI Reagent (Sigma-Aldrich, cat# T9424) added directly to each tissue culture dish with 1-bromo-3-chloropropane (Sigma-Aldrich, cat# B9673) added to the homogenous lysate according to manufacturer's instructions. The RNA pellet was dried and stored at −80°C until shipped on dry-ice for quantitative PCR. RNA concentration and purity was determined (quality control data available at https://osf.io/yh8z9/ ). Total RNA was reverse transcribed into cDNA using a First-Strand cDNA Synthesis Kit (Sigma-Aldrich, cat# GE27-9261-01) according to manufacturer's instructions. Reactions consisted of cDNA (2 µl of 10 ng/µl), qPCR Mastermix, ultrapure water, and primers (forward and reverse primers are listed in the Registered Report Protocol 3; *Khan et al., 2015*). qRT-PCR reactions were performed in technical triplicate with QuantiTect Sybr Green PCR Kit (Qiagen, cat# 204141) according to manufacturer's instructions. PCR cycling conditions were: one cycle: 95°C for 30 s – 40 cycles: [95°C for 10 s, 60°C for 30 s] – one cycle: 95°C for 30 s, 65–95°C (ramp 0.5°C every 5 s) using a BioRad CFX96 qPCR system (Hercules, California) and CFX Manager software (RRID:SCR_017251), version 3.1.1517.0823. Negative controls containing no cDNA template were included. It was discovered the melting curves for *PTEN* and *PTENP1* were inconsistent and required repeating measuring. Relative expression levels were determined using the ΔΔCt method. Data files are available at https://osf.io/6zbdt/.

### siRNA transfection and western blot

DU145 cells were seeded at $3.75 \times 10^5$ cells per well in a six well plate and transfected with siRNAs as described above in 'siRNA transfection and crystal violet proliferation assay'. 48 hr after transfection, cells were prepared in lysis buffer (50 mM Tris, pH 8.0, 150 mM NaCl, 1% NP-40, 1 mM EDTA, 1 mM $MgCl_2$, 1 mM ß-glycerophosphate, 1 mM NaF, and 1 mM $Na_3VO_4$), supplemented with protease inhibitors (Sigma-Aldrich, cat# P8340) at manufacturer recommended concentrations. Lysed cells were incubated on ice for 30 min, gently sonicated for 3 to 4 bursts for 5 to 10 s, and centrifuged at 10,000x*g* for 10 min at 4°C before protein concentration of supernatant was quantified using a Bradford assay following manufacturer's instructions. Lysate samples (2.5 µg was used after optimization studies) were separated by 4–12% Tris Glycine SDS-PAGE gel electrophoresis with 5 µl of an ECL marker (Sigma-Aldrich, cat# GERPN810) and then transferred to a nitrocellulose membrane as described in the Registered Report (Protocol 4; *Khan et al., 2015*), except membranes were not washed with methanol prior to transfer. Transfer was confirmed by Ponceau S staining and membranes were blocked with 5% non-fat dry milk in 1X TBS with 0.1% Tween-20 (TBST) overnight at 4°C. Membranes were probed with the following primary antibodies diluted in 5% non-fat dry milk in TBST for 2 hr at room temperature: rabbit anti-PTEN [clone 138G5] (Cell Signaling Technology, cat# 9559, RRID:AB_390810), 1:1000 dilution; mouse anti-HSP90 [clone 68] (BD Biosciences, cat# 610419, RRID:AB_397798), 1:1000 dilution. Membranes were washed with TBST and incubated with secondary antibody diluted in 5% non-fat dry milk in TBST: HRP-conjugated goat anti-rabbit (Cell Signaling Technology, cat# 7074, RRID:AB_2099233), 1:20,000 dilution; HRP-conjugated horse anti-mouse (Cell Signaling Technology, cat# 7076, RRID:AB_330924), 1:20,000 dilution. Membranes were washed with TBST and incubated with ECL reagent to visualize signals. Scanned Western blots were quantified using ImageJ software (RRID:SCR_003070), version 1.50a (*Schneider et al., 2012*). Additional methods and data, including full Western blot images, are available at https://osf.io/re87y/.

### PTEN 3'UTR plasmid generation

To generate pCMV-*PTEN*-3'UTR (deposited in Addgene, plasmid# 97204; RRID:Addgene_97204), *PTEN* 3'UTR was amplified from HeLa genomic DNA (New England BioLabs, cat# N4006) by PCR using KAPA Bio's HiFi Hotstart ReadyMix (Kapa Biosystems, cat# KM2605) and the following primers

(BamH1 forward primer: 5'-GGATCCTAGAGGAGCCGTCAAATCCA-3'; XhoI reverse primer: 5'-C TCGAGTGGACATCTGATTGGGATGA-3'). PCR cycling conditions were: one cycle: 98°C for 30 s – 35 cycles: 98°C for 15 s, 64°C for 30 s, 72°C for 3 min 30 s – one cycle: 72°C for 2 min. Following multiple unsuccessful attempts to clone the PCR product (~3.3 kb) into the pCMV vector (Agilent Technologies, cat# 240071) (direct, TOPO TA, Gibson assembly), we were successful by cloning partial fragments. The full-length *PTEN* 3'UTR PCR product was able to be broken into three fragments with the following primers and cycling conditions. PCR primers and parameters for CMV-*PTEN*-3'UTR minus middle SpeI-BstXI fragment: [CMV-*PTEN*-3'UTR minus SpeI/BstXI forward primer: 5'-GAAATTTGGTGTCTTCAAATTATACCTTCAC-3'; CMV-*PTEN*-3'UTR minus SpeI/BstXI reverse primer: 5'-TTGAAAACTAGTAAAATAAGTGTAAGTTGTTGACTG-3']; one cycle: 98°C for 30 s – 35 cycles: 98°C for 15 s, 58°C for 30 s, 72°C for 5 min – 1 cycle: 72°C for 2 min. PCR primers and parameters for *PTEN*-3'UTR SpeI-BstXI fragment: [*PTEN*-3'UTR SpeI-BstXI forward primer: 5'-CACTTA TTTTACTAGTTTTCAATCATAATACCTG-3'; *PTEN*-3'UTR SpeI-BstXI reverse primer: 5'-TTGAAGA-CACCAAATTTCTGGAAAAAAAAACC-3']; 1 cycle: 98°C for 30 s – 35 cycles: 98°C for 15 s, 58°C for 30 s, 72°C for 1 min – 1 cycle: 72°C for 2 min. PCR products were gel purified and a 1:3 vector to insert ratio (0.02 pmoles: 0.06 pmoles) was used for the Gibson Assembly reactions (New England BioLabs, cat# E5510S) according to manufacturer's instructions (50°C, 15 min). The assembly reaction was transformed into chemically competent *E. coli* (New England BioLabs, cat# C3040I) according to manufacturer's instructions, except 1 hr phenotypic expression and plating temperatures were performed at 30°C. A positive clone was verified by sequencing. Additional methods are available at https://osf.io/9stbg/.

## PTEN 3'UTR transfection and quantitative PCR

DU145 cells were seeded at $3.5 \times 10^5$ cells per well in 60 mm plates and grown overnight. The next day cells were washed once with phosphate buffered saline (Sigma-Aldrich, cat# D8537) before transfection using Effectene (Qiagen, cat# 301425) according to manufacturer's instructions, or left untransfected. 1 μg plasmid (pCMV (empty vector) or pCMV-*PTEN*-3'UTR) was mixed with EC to a final volume of 150 μl followed by an addition of 8 μl of Enhancer, mixed for 1 s and then incubated at 25°C for 5 min. 25 μl of Effectene transfection reagent was added to the DNA-Enhancer sample and vortexed for 10 s before incubation at room temperature for 10 min. ~0.6 ml of growth medium was added to the transfection complex and mixed before added dropwise to dishes containing cells. 6 hr after transfection growth medium was replaced. 24 hr after transfection, total RNA was extracted and qPCR was performed as described above in 'siRNA transfection and quantitative PCR'. Data files are available at https://osf.io/9etxj/.

## PTEN 3'UTR transfection and crystal violet proliferation assay

DU145 cells were seeded and transfected as described above in '*PTEN* 3'UTR transfection and quantitative PCR' above. 6 hr after transfection cells were plated at 8,000 cell per well (in triplicate) in 12 well plates (enough for 6 days of measurements) with 2 ml of growth medium. Starting the day after plating (designated day 0), every 24 hr the crystal violet assay was performed as described in the Registered Report. Absorbance was recorded at 590 nm using SpectramaxPlus (Molecular Devices, Serial # P02528) and SoftMax Pro data acquisition and analysis software (RRID:SCR_014240), version 4.6. For each independent biological repeat, average absorbance ($OD_{590}$) for each condition was normalized by dividing the average absorbance of each day to the average absorbance for day 0 to calculate relative cell number. Area under the curve (AUC) was calculated for each condition of each biological repeat. Raw data files are available at https://osf.io/qf7xa/.

### Statistical analysis

Statistical analysis was performed with R software (RRID:SCR_001905), version 3.5.1 (*R Development Core Team, 2018*). All data, csv files, and analysis scripts are available on the OSF (https://osf.io/yyqas/). Confirmatory statistical analysis was pre-registered (https://osf.io/2evzy) before the experimental work began as outlined in the Registered Report (*Khan et al., 2015*). Data were checked to ensure assumptions of statistical tests were met. When described in the results, the Bonferroni correction, to account for multiple tests, was applied to the alpha error or the *p*-value. The Bonferroni corrected value was determined by dividing the uncorrected value (0.05) by the

number of tests performed. Although the Bonferroni method is conservative, it was accounted for in the power calculations to ensure sample size was sufficient. A meta-analysis of a common original and replication effect size was performed with a random effects model and the *metafor* R package (*Viechtbauer, 2010*) (https://osf.io/9yh6p/). The original study data was extracted a priori from the published figures by estimating the value reported. The extracted data was published in the Registered Report (*Khan et al., 2015*) and was used in the power calculations to determine the sample size for this study.

## Data availability

Additional detailed experimental notes, data, and analysis are available on OSF (RRID:SCR_003238) (https://osf.io/yyqas/; *Pandya et al., 2019*). This includes the R Markdown file (https://osf.io/v3cag/) that was used to compose this manuscript, which is a reproducible document linking the results in the article directly to the data and code that produced them (*Hartgerink, 2017*).

## Deviations from registered report

The first protocol of the Registered Report was conducted, but it was identified after results were obtained that the si-miR-19b and si-miR-20a reagents listed in the Registered Report and used in the experiment were incorrect and thus are not included in this manuscript. The catalog numbers listed were for microRNA hairpin inhibitors not microRNA mimics. The original study does not list the catalog numbers and only lists the reagents as si-miR-19b or si-miR-20a. The expected result of reduced PTEN levels following overexpression of *miR-19b* or *miR-20a* that has been observed previously (e.g., *Luo et al., 2013*; *Poliseno et al., 2010*; *Tian et al., 2013*) was not observed, which lead to the reevaluation of the reagent sources. Results obtained from using the inhibitors are available at https://osf.io/fjdtn/. The fifth and sixth protocol of the Registered Report (results presented in *Figures 4*,*5*) used a plasmid that overexpressed *PTEN* 3'UTR that was generated as part of this replication attempt while the Registered Report indicated the plasmid would be shared by the original lab. The reason to remake the plasmid was because the original lab did not respond to our multiple follow up requests to share the plasmid with us following acceptance of the Registered Report. Details of how the plasmid was made are described in the '*PTEN 3'UTR plasmid generation*' section of the Materials and methods and the plasmid was deposited in Addgene (plasmid# 97204; RRID: Addgene_97204) for the research community to access. Additional materials and instrumentation not listed in the Registered Report, but needed during experimentation are also listed above.

## Acknowledgements

The Reproducibility Project: Cancer Biology would like to thank Laura Poliseno (Institute of Clinical Physiology-Italian National Research Council), Leonardo Salmena (University of Toronto), and Pier Paolo Pandolfi (Beth Israel Deaconess Cancer Center) for sharing critical protocol information during preparation of the Registered Report. We would also like to thank ProNovus Bioscience, LLC for assistance in generating the *PTEN* 3'UTR plasmid and Dale Cowley and Kumar Pandya, TransViragen, Inc, for conducting the knockdown experiments (specifically the cell growth assay, Western blot, and preparation of RNA for qPCR) as well as feedback on this manuscript. We thank the following companies for generously donating reagents to the Reproducibility Project: Cancer Biology; American Type and Tissue Collection (ATCC), Applied Biological Materials, BioLegend, Charles River Laboratories, Corning Incorporated, DDC Medical, EMD Millipore, Harlan Laboratories, LI-COR Biosciences, Mirus Bio, Novus Biologicals, Sigma-Aldrich, and System Biosciences (SBI).

## Additional information

### Group author details

**Reproducibility Project: Cancer Biology**
**Elizabeth Iorns**: Science Exchange, Palo Alto, United States; **Rachel Tsui**: Science Exchange, Palo Alto, United States; **Alexandria Denis**: Center for Open Science, Charlottesville, United States;

**Nicole Perfito**: Science Exchange, Palo Alto, United States; **Timothy M Errington**: Center for Open Science, Charlottesville, United States

## Competing interests

John Kerwin: University of Maryland College Park is a Science Exchange associated lab. Israr Khan: Alamo Laboratories Inc is a Science Exchange associated lab. Reproducibility Project: Cancer Biology: EI, RT, NP: Employed by and hold shares in Science Exchange Inc.The other authors declare that no competing interests exist.

## Funding

| Funder | Author |
|--------|--------|
| Laura and John Arnold Foundation | Reproducibility Project: Cancer Biology |

The funder had no role in study design, data collection and interpretation, or the decision to submit the work for publication.

## Author contributions

John Kerwin, Israr Khan, Data curation, Methodology, Writing - review and editing; Reproducibility Project: Cancer Biology, Conception and design, Analysis and interpretation of data, Writing - review and editing, analysis

## Author ORCIDs

Alexandria Denis (iD) http://orcid.org/0000-0002-1210-2309
Timothy M Errington (iD) http://orcid.org/0000-0002-4959-5143

## Decision letter and Author response

Decision letter https://doi.org/10.7554/eLife.51019.sa1
Author response https://doi.org/10.7554/eLife.51019.sa2

# Additional files

## Supplementary files

• Transparent reporting form

## Data availability

Additional detailed experimental notes, data, and analysis are available on OSF (RRID:SCR_003238) (https://osf.io/yyqas/; Pandya et al., 2018). This includes the R Markdown file (https://osf.io/v3cag/) that was used to compose this manuscript, which is a reproducible document linking the results in the article directly to the data and code that produced them (Hartgerink, 2017).

The following dataset was generated:

| Author(s) | Year | Dataset title | Dataset URL | Database and Identifier |
|-----------|------|---------------|-------------|-------------------------|
| Pandya K, Kerwin J, Cowley D, Khan I, Iorns E, Tsui R, Denis A, Perfito N, Errington TM | 2019 | Study 1: Replication of Poliseno et al., 2010 (Nature) | https://dx.doi.org/10.17605/OSF.IO/YYQAS | Open Science Framework, 10.17605/OSF.IO/YYQAS |

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
