## [Decision Letter]

**Decision letter after peer review:**

Thank you for submitting your article "Replication Study: A coding-independent function of gene and pseudogene mRNAs regulates tumour biology" for consideration by *eLife*. Your article has been reviewed by three peer reviewers, one of whom is a member of our Board of Reviewing Editors, and the evaluation has been overseen by Charles Sawyers as the Senior Editor. The reviewers have opted to remain anonymous.

The Reviewing Editor has drafted this decision to help you prepare a revised submission. We have appended the comments from the three reviewers to provide context for our seven points listed under "Essential revisions."

Summary:

The manuscript by Kerwin and colleagues is reporting on data generated during the Reproducibility Project related to a paper published by Pandolfi's group (Poliseno et al., 2010) describing an RNA interaction function for the PTEN pseudogene. This paper was seen as a major advancement on the field of non-codingRNAs at the time of publication (see the comments in Nature 2010 and Nat Rev Cancer 2010). This publication inspired also a large effort by Chinnaiyan's group, which reported the translational landscape of expressed pseudogenes in human cancers. Therefore, the Poliseno et al. publication had a major impact on the field of non-codingRNAs independent of Dr. Pandolfi's own research.

There is also an error in the Registered Report (wrong reagent), but this is acknowledged in the subsection "Deviations from Registered Report" and these data should not be presented – this section of the Replication Study is therefore missing. This is unfortunate, but there is no major problem since the data will not be presented.

The Replication Report indicates that several biological assays (e.g. proliferation) reported by Poliseno et al., 2010 were reproduced, but many of the molecular assays were not reproduced.

Essential revisions:

For clarity, the individual reviews are appended to this decision letter. It is recommended that these reviews are used to assist the preparation of a revised manuscript by addressing all the points that are raised. In particular, the following revisions should be fully addressed.

1) The Abstract appears to have been prepared using a template and does not adequately convey the results and conclusion of the study.

2) Proper citation of all the relevant literature that reports replication of the original study is required (see attached reviews).

3) Figure 1. It appears that transfection alone has a phenotype – is this a technical flaw in the performance of these experiments? The failure of si-PTEN to cause increased proliferation is surprising – this also raises questions concerning a possible technical flaw in these studies.

4) Figure 2. Please comment on the very large variability in the data obtained and the different normalization standards (36B4 vs Actin) used compared with Poliseno et al., 2010. Does this compromise the interpretation of these data – if so, this should be stated.

5) Figure 3. Please confirm the quantitation and statistical analysis (see attached reviews).

6) Figure 4. Why was the PTEN3'UTR plasmid employed different than the plasmid noted in the Registered Report (the one used by Poliseno et al., 2010)? What criteria were used to validate the plasmid that was used?

7) A major conclusion of this study is that many of the molecular studies of Poliseno et al., 2010 were not reproduced in this specific analysis. This is a conclusion supported by the data presented. However, this conclusion should be reported in the context of noting that reproduction has been reported in the literature (including citations). The failure to replicate could be due to the particular experimental conditions (e.g. state of cell confluence, transfection-related toxicity, etc), perhaps the use of different normalization standards, the large variability of some of the data reported here that might prevent the detection of significant differences, and differences in technical ability between the personnel doing the studies. These (and other) issues should be noted as caveats to the overall study reported here.

Reviewer #1:

This appears to be a properly executed Registered Report. The Introduction summarizes the target paper and states the history of the field with supporting evidence. The replication study itself indicates that the more molecular analyses (expression data) were poorly reproduced, but some of the more descriptive data (growth assays) were better reproduced. I did not see a major flaw in the reproduction study. Moreover, the last paragraph of the report notes a number of caveats that need to be applied to the interpretation of this study in comparison to the reference study. I have no recommendations for revision.

Reviewer #2:

In this manuscript, Kerwin and colleagues have attempted to replicate Figure 2F-H and Figure 4A of the paper "A coding-independent function of gene and pseudogene mRNAs regulates tumor biology" (Poliseno et al., 2010). Overall, the results go in the same direction as the ones reported in the original paper (PTEN and PTENP1 are shown to be coregulated), but in some instances statistical significance is not reached and, in our opinion, this is at least in part due to the poor quality of the data.

The manuscript is less than convincing, for the reasons listed below:

- The ONE western blot that is shown is quite poor in quality, which is disappointing given the excellent quality of PTEN antibodies. This should be repeated.

- Standard deviations in qRT-PCR are quite large throughout paper.

- Aim of each experiment is unclear, i.e. the main difference in expression levels/biological properties that each experiment wants to reproduce is never clearly stated. In the absence of any explanatory framework, the description of experimental results appears as a tedious list of statistical analyses.

- Finally, the subsection "Deviation from registered report" is worrisome. The authors report that they could not reproduce Figure 1D of the original paper (miR-19b and miR-20a overexpression causes a decrease in both PTEN and PTENP1 RNA levels) because they realized they had used microRNA inhibitors instead of miRNA mimics, which means they had inhibited the microRNAs rather than overexpressing them! We recommend either removing this from manuscript or performing the replicating experiment as planned.

After reading this manuscript, we are even more convinced that the best validation of published data is the publication, by the original group as well as by independent groups, of more data that replicate the original findings. The sponging interaction between PTENP1 and PTEN, which was originally published by Poliseno et al., 2010, has been by now reproduced in 15 papers/studies, all by different groups. We have one in preparation ourselves, in which we confirm the original data obtained using RNA interference and we show that they hold true using cutting-edge CRISPR-based technologies.

As a final note, we find it puzzling that the authors themselves reference the literature (specifically, to the paper they intend to replicate as well as to others) to realize their mistake about the reproduction attempt of Figure 1D (subsection "Deviation from registered report": The expected result of reduced PTEN levels following overexpression of miR-19b or miR-20a... has been observed previously (e.g. Luo et al., 2013; Poliseno et al., 2010; Tian et al., 2013; Wu et al., 2014)). Why in this instance did they choose to not replicate?

References included in the manuscript

- The list of papers that confirm PTENP1 as sponging partner of PTEN needs to be extended and updated. The following references need to be added:

1) Johnsson et al., 2013.

2) Yu et al., 2014.

3) Li et al., 2017.

4) Qian et al., 2017.

5) Shi, Tang and Su, 2017.

6) Chen et al., 2018.

7) Gao et al., 2019.

8) Johnson et al., 2019.

9) Wang et al., 2016.

10) Lai et al., 2019.

- Ynstead et al., 2018: the results described in this paper are not reported correctly.

- The paper Wu et al., 2014 does not demonstrate directly that PTEN is targeted by miR-19b or 20a. So we are not sure why it is quoted in subsection "Deviations from Registered Report".

Figure 1 (replication of Figure 2F of the original paper)

- There is a large difference btw untransfected and transfected cells. Did the authors check for transfection toxicity?

- si-PTEN should speed up cell growth. Why is this not seen?

Figure 2 (replication of Figure 2G of the original paper)

- The pattern of results is similar to the one reported in the Nature paper, however extremely large error bars, which prevent differences from reaching statistical significance, suggest that there was large variability in repeats. This in turn is suggestive of experimental consistency issues.

- The use of 36B4 instead of Actin as internal control represents a difference compared to the experimental protocol followed in the original paper.

Figure 3 (replication of Figure 2H of the original paper)

- The quality of western blot is very bad. This is surprising given the robustness and reliability of PTEN western antibodies in general. Should be repeated or better western displayed.

- HSP90 is overexposed. Quantitation may be compromised.

- It is not clear how quantitation was performed as results appear to be binned, whereas quantitation should be continuous ratios btw PTEN/HSP90.

- The authors claim that the decrease in PTEN levels upon the transfection of si-PTEN and si-PTEN/PTENP1 is statistically significant, while that observed upon the transfection of si-PTENP1 is not. The latter is the main point that the experiment wants to make and clearly 1 replicate is skewing this data. Saying that there is no reproducibility just because 1 replicate is off compared to the other 4 is against common sense. Data interpretation is essential to the work of each scientist, as it is the one that allows to formulate the hypotheses that will guide the design of the experiments to come.

Figure 4 (replication of Figure 4A of the original paper)

- Did the authors determine if their plasmid can successfully overexpress PTEN3'UTR? This needs to be determined.

- Why did the authors make this plasmid themselves, instead of using the one used in the original paper, as indicated in the Registered Report?

- This result was replicated consistently in Tay et al., 2011 from same group (PMID: 22000013).

Reviewer #3:

The manuscript by Kerwin and colleagues is reporting on data generated during the Reproducibility Project related to an important paper published by Pandolfi's group (Poliseno et al., 2010) describing an RNA interaction function for the PTEN pseudogene.

The same group of authors published in 2015 a registered report on how the confirmation workflow will be done that I reviewed and approved myself. This report was twice corrected on-line, once technically and once reporting the removal of the first two authors of the study with their inclusion in acknowledgements.

This Pandolfi group publication was seen as a major advancement on the field of non-codingRNAs at the time of publication (see the comments in Nature 2010 and Nat Rev Cancer 2010). This publication inspired also a large effort by Chinnaiyan's group who produced the translational landscape of expressed pseudogenes in human cancers. Therefore, the Poliseno et al., publication had, in fact, a major impact on the field of non-codingRNAs independent of Dr. Pandolfi's own research.

This is the basis of the selection of the paper to be reproduced, making consequently the report by Kerwin and colleagues more interesting regarding data reproducibility for a large spectrum of readers independently on the specific focus of their research. It has to be appreciated the effort and work and time commitment by Kerwin and colleagues on this.

1) Regarding the Abstract, this report is from a series of reports on the same theme and has the wording pattern from other similar reports published from the same group. What is surprisingly missing from all these abstracts is the conclusion sentence that is a standard for scientific publications. Please add a conclusion: did this report generally confirmed or not the data from Poliseno et al., 2010? I would recommend also for future studies on other publications to have a conclusive statement in the Abstract.

2) In Figure 1, I would add a second panel with the actual levels of downregulation of PTEN, PTENP1 and PTEN/PTENP1, and also compare these levels with the ones reported by Poliseno et al. If these are not reported in Nature paper from 2010, then this information has to be requested from Pandolfi's group. It is essential, as various levels of downregulation can give various phenotype effects.

3) The lack of replication of reported data in subsection "Quantitative PCR following transfected with siRNA against PTEN and/or PTENP1", meaning the lack of significant variation of PTEN mRNA levels following PTENP1 knockdown and vice versa are problematic, as these are related to the core finding of Poliseno's et al. paper... Starting from simple technical points, were the Î² actin levels by qRT-PCR in the replication study comparable with the ones from Poliseno's study? If not, then this can be an explanation. Second, was the confluence of cells in the experiment by Kerwin and colleagues' for this particular experiment similar with the cell confluence levels used by Poliseno et al.? It is already reported that cell confluence significantly modifies the levels of expression of ncRNAs, see please a seminal paper by Mandel's group (Hwang et al., 2009). As this info is not reported in Nature paper and also is not used to be reported in publications, direct communication with the authors from Poliso et al., paper is important. Generally, as this is such an essential point for the ceRNA theory, the data have to be compared point by point between the two groups.

4) For Figure 3B, please add the quantification results of PTEN protein levels. Same question as above: was the confluence of cells in the experiment by Kerwin and colleagues' fir this particular experiment similar with the cell confluence levels for Poliseno et al.?

5) The lack of replication of reported data in subsection "Quantitative PCR following PTEN 3` UTR transfection", meaning the lack of effect of exogenous PTEN-3UTR on the expression of PTENP1 is also problematic regarding Poliseno's report...Again, the levels of normalizers for the two sets of experiments, Kerwin and colleagues versus Poliseno et al., as well as cell confluence, the cell passage used for these studies, have to be compared side by side. If not possible, I suggest the inclusion of this points specifically in the discussion, as a knowledgeable reader can't draw conclusions without such information.

---

## [Author Response]

Essential revisions:For clarity, the individual reviews are appended to this decision letter. It is recommended that these reviews are used to assist the preparation of a revised manuscript by addressing all the points that are raised. In particular, the following revisions should be fully addressed.1) The Abstract appears to have been prepared using a template and does not adequately convey the results and conclusion of the study.

We've revised the abstract to include additional information, but we did not include a statement regarding whether this report generally confirmed (or not) the data from Poliseno et al., 2010, similar to our other reports. *eLife* has been publishing editors' summaries from their perspective of whether the results were generally confirmed (or not) and since there is no agreement of what it means to confirm (or not) an original study we prefer to remain silent on making this claim in the individual replication reports. Finally, we will publish a final paper that summarizes the collective evidence of all the replications, which will provide a better platform for discussing this important and complex issue.

2) Proper citation of all the relevant literature that reports replication of the original study is required (see attached reviews).

We have included additional literature as suggested by reviewer #2.

3) Figure 1. It appears that transfection alone has a phenotype – is this a technical flaw in the performance of these experiments? The failure of si-PTEN to cause increased proliferation is surprising – this also raises questions concerning a possible technical flaw in these studies.

We agree it is possible that transfection toxicity could be a variable at play in both the original and replication study, which the inclusion of the untransfected condition in the replication surfaces, a condition not reported in the original study. Visualization of the cells by microscopy revealed the transfected and untransfected cells were similar to each other. Another important difference is the replication showed a growth rate much higher in the control condition than in the original study, which reported little to no growth in control cells over the time course, but which is still lower than the untransfected condition. The impact of lipid-based transfection on cellular processes has been documented, so this is also not unexpected and illustrates the need for untransfected cells for proper interpretation. We have included these points and a further discussion on this point in the revised manuscript. We also agree the level of increased cell growth in the siPTEN condition compared to control was expected to be higher than observed in this replication attempt and have included this in the revised manuscript.

4) Figure 2. Please comment on the very large variability in the data obtained and the different normalization standards (36B4 vs Actin) used compared with Poliseno et al., 2010. Does this compromise the interpretation of these data – if so, this should be stated.

The variation between the biological repeats suggests variation in biology as well as any technical variation associated with the plating and transfection step of the experiment. We have included a discussion of the impact this has on interpretation of these data. The use of 36B4 as an additional internal control was asked of us during peer review of the Registered Report (https://doi.org/10.7554/eLife.08245.002). This point has been included in the revised manuscript.

5) Figure 3. Please confirm the quantitation and statistical analysis (see attached reviews).

We have addressed the comments regarding quantitation and statistic in the attached reviews.

6) Figure 4. Why was the PTEN3'UTR plasmid employed different than the plasmid noted in the Registered Report (the one used by Poliseno et al., 2010)? What criteria were used to validate the plasmid that was used?

We have included the reason in the subsection "Deviations from Registered Report". In short, the original lab that offered to share the plasmid did not respond to our multiple follow up requests to share the plasmid with us following acceptance of the Registered Report. We also agree it is possible that although the plasmid was verified to be correct by sequencing that PTEN 3'UTR was not expressed (or not expressed to the same level as the original study). We have raised this point in the revised manuscript as well as how differences in the level of expression could impact the results compared to the original study.

7) A major conclusion of this study is that many of the molecular studies of Poliseno et al., 2010 were not reproduced in this specific analysis. This is a conclusion supported by the data presented. However, this conclusion should be reported in the context of noting that reproduction has been reported in the literature (including citations). The failure to replicate could be due to the particular experimental conditions (e.g. state of cell confluence, transfection-related toxicity, etc), perhaps the use of different normalization standards, the large variability of some of the data reported here that might prevent the detection of significant differences, and differences in technical ability between the personnel doing the studies. These (and other) issues should be noted as caveats to the overall study reported here.

We have revised the manuscript throughout to discuss these caveats and have addressed the specific reviewer comments related to these.

Reviewer #1:This appears to be a properly executed Registered Report. The Introduction summarizes the target paper and states the history of the field with supporting evidence. The replication study itself indicates that the more molecular analyses (expression data) were poorly reproduced, but some of the more descriptive data (growth assays) were better reproduced. I did not see a major flaw in the reproduction study. Moreover, the last paragraph of the report notes a number of caveats that need to be applied to the interpretation of this study in comparison to the reference study. I have no recommendations for revision.

Thank you.

Reviewer #2:In this manuscript, Kerwin and colleagues have attempted to replicate Figure 2F-H and Figure 4A of the paper "A coding-independent function of gene and pseudogene mRNAs regulates tumor biology" (Poliseno et al., 2010). Overall, the results go in the same direction as the ones reported in the original paper (PTEN and PTENP1 are shown to be coregulated), but in some instances statistical significance is not reached and, in our opinion, this is at least in part due to the poor quality of the data.The manuscript is less than convincing, for the reasons listed below:- The ONE western blot that is shown is quite poor in quality, which is disappointing given the excellent quality of PTEN antibodies. This should be repeated.

We used the same PTEN antibody as the original study and optimized conditions based on the original protocol as stated in the Materials and methods section. We revised Figure 3 to include two exposures to help facilitate detection of PTEN. Further, all western blot images and exposures are available at https://osf.io/n6vn3/, which will be made public upon publication.

- Standard deviations in qRT-PCR are quite large throughout paper.

Further discussion on possible reasons for this are also included in the revised manuscript.

- Aim of each experiment is unclear, i.e. the main difference in expression levels/biological properties that each experiment wants to reproduce is never clearly stated. In the absence of any explanatory framework, the description of experimental results appears as a tedious list of statistical analyses.

The aim of each experiment is briefly stated at the beginning of each experiment section. However, we have revised the manuscript to better address the dependent and independent variables under investigation.

- Finally, the subsection "Deviation from registered report" is worrisome. The authors report that they could not reproduce Figure 1D of the original paper (miR-19b and miR-20a overexpression causes a decrease in both PTEN and PTENP1 RNA levels) because they realized they had used microRNA inhibitors instead of miRNA mimics, which means they had inhibited the microRNAs rather than overexpressing them! We recommend either removing this from manuscript or performing the replicating experiment as planned.

This experiment is not included in this manuscript as stated in the subsection "Deviations from Registered Report". The error occurred somewhere during the preparation of the Registered Report and was not caught during peer review, however because of the unexpected result we confirmed all details and identified this error and thus are not including the data/results in this manuscript.

After reading this manuscript, we are even more convinced that the best validation of published data is the publication, by the original group as well as by independent groups, of more data that replicate the original findings. The sponging interaction between PTENP1 and PTEN, which was originally published by Poliseno et al., 2010, has been by now reproduced in 15 papers/studies, all by different groups. We have one in preparation ourselves, in which we confirm the original data obtained using RNA interference and we show that they hold true using cutting-edge CRISPR-based technologies.As a final note, we find it puzzling that the authors themselves reference the literature (specifically, to the paper they intend to replicate as well as to others) to realize their mistake about the reproduction attempt of Figure 1D (subsection "Deviation from registered report": The expected result of reduced PTEN levels following overexpression of miR-19b or miR-20a... has been observed previously (e.g. Luo et al., 2013; Poliseno et al., 2010; Tian et al., 2013; Wu et al., 2014)). Why in this instance did they choose to not replicate?

We do not understand the final question "Why in this instance did they choose to not replicate?". Is the suggestion that because we cite the data from the original study, as well as other studies, we should not have chosen to replicate this specific experiment? The decisions over what papers and experiments to replicate were described in our overview paper of the project (https://elifesciences.org/articles/04333) and for this specific replication were discussed during peer review of the Registered Report (https://elifesciences.org/articles/08245).

References included in the manuscript- The list of papers that confirm PTENP1 as sponging partner of PTEN needs to be extended and updated. The following references need to be added:1) Johnsson et al., 2013.2) Yu et al., 2014.3) Li et al., 2017.4) Qian et al., 2017.5) Shi, Tang and Su, 2017.6) Chen et al., 2018.7) Gao et al., 2019.8) Johnson et al., 2019.9) Wang et al., 2016.10) Lai et al., 2019.

We have included additional references in the revised manuscript in addition to the references that were already included on the role of PTEN/PTENP1 in cancer.

- Ynstead et al., 2018: The results described in this paper are not reported correctly.

Thank you for catching this error. We have revised the sentence referencing the results of Yndestad et al., 2018 to reflect that *PTEN* expression was decreased, not increased, in MCF7 cells when *PTENP1* was upregulated.

- The paper Wu et al., 2014 does not demonstrate directly that PTEN is targeted by miR-19b or 20a. So we are not sure why it is quoted in subsection "Deviations from Registered Report".

We have removed Wu et al., 2014 in the revised manuscript.

Figure 1 (replication of Figure 2F of the original paper)- There is a large difference btw untransfected and transfected cells. Did the authors check for transfection toxicity?

We agree it is possible that transfection toxicity could be a variable at play in both the original and replication study, which the inclusion of the untransfected condition in the replication surfaces, a condition not reported in the original study. Visualization of the cells by microscopy revealed the transfected and untransfected cells were similar to each other. Another important difference is the replication showed a growth rate much higher in the control condition than in the original study, which reported little to no growth in control cells over the time course, but which is still lower than the untransfected condition. The impact of lipid-based transfection on cellular processes has been documented, so this is also not unexpected and illustrates the need for untransfected cells for proper interpretation. We have included these points and a further discussion on this point in the revised manuscript.

- si-PTEN should speed up cell growth. Why is this not seen?

We agree the level of increased cell growth in the siPTEN condition compared to control was expected to be higher than observed in this replication attempt and have included this in the revised manuscript.

Figure 2 (replication of Figure 2G of the original paper)- The pattern of results is similar to the one reported in the Nature paper, however extremely large error bars, which prevent differences from reaching statistical significance, suggest that there was large variability in repeats. This in turn is suggestive of experimental consistency issues.

The variation between the biological repeats suggests variation in biology as well as any technical variation associated with the plating and transfection step of the experiment. We have included a discussion of the impact this has on detecting statistical significance.

- The use of 36B4 instead of Actin as internal control represents a difference compared to the experimental protocol followed in the original paper.

The use of 36B4 as an additional internal control was asked of us during peer review of the Registered Report (https://doi.org/10.7554/eLife.08245.002). This point has been included in the revised manuscript.

Figure 3 (replication of Figure 2H of the original paper)- The quality of western blot is very bad. This is surprising given the robustness and reliability of PTEN western antibodies in general. Should be repeated or better western displayed.

We used the same PTEN antibody as the original study and optimized conditions based on the original protocol as stated in the Materials and methods. We revised Figure 3 to include two exposures to help facilitate detection of PTEN. Further, all western blot images and exposures are available at https://osf.io/n6vn3/, which will be made public upon publication.

- HSP90 is overexposed. Quantitation may be compromised.

Multiple exposures of PTEN and HSP90 were taken and used in quantitation.

- It is not clear how quantitation was performed as results appear to be binned, whereas quantitation should be continuous ratios btw PTEN/HSP90.

Additional details on quantitation has been added to the revised manuscript. Results are not binned, but are on a continuous scale, relative to siLuc (which is arbitrarily set to 1). We also revised the figure to create more separation in the individual data points where possible, but where values are similar they appear binned even though they are not.

- The authors claim that the decrease in PTEN levels upon the transfection of si-PTEN and si-PTEN/PTENP1 is statistically significant, while that observed upon the transfection of si-PTENP1 is not. The latter is the main point that the experiment wants to make and clearly 1 replicate is skewing this data. Saying that there is no reproducibility just because 1 replicate is off compared to the other 4 is against common sense. Data interpretation is essential to the work of each scientist, as it is the one that allows to formulate the hypotheses that will guide the design of the experiments to come.

There is no statement made in the manuscript about 'no reproducibility'. The reviewer is correct the main test of the experiment was not statistically significant, but this comment appears to equate 'statistically significant' with 'reproduced' and vice versa. Failing to reject the null hypothesis does not mean we accept the null hypothesis, only that there is insufficient evidence at the predefined level of significance and power (sample size) to reject the null hypothesis. We also disagree with the comment that '1 replicate is skewing the data'. Unlike the original study that reported an N of 1, the variation of multiple biological repeats are shown in this replication study and necessary for appropriate interpretation.

Figure 4 (replication of Figure 4A of the original paper)- Did the authors determine if their plasmid can successfully overexpress PTEN3'UTR? This needs to be determined.

Thank you for raising this point. It is possible that although the plasmid was verified to be correct by sequencing that PTEN 3'UTR was not expressed. We have raised this point in the revised manuscript as well as how differences in the level of expression could impact the results compared to the original study.

- Why did the authors make this plasmid themselves, instead of using the one used in the original paper, as indicated in the Registered Report?

We have included the reason in the subsection "Deviations from Registered Report". In short, the original lab that offered to share the plasmid did not respond to our multiple follow up requests to share the plasmid with us following acceptance of the Registered Report.

- This result was replicated consistently in Tay et al., 2011 from same group (PMID: 22000013).

This specific experiment (Figure 4A of the original paper and Figure 4 of this manuscript) that examined expression of *PTENP1* following overexpression of the *PTEN* 3'UTR, does not seem to be reported in Tay et al., 2011. Although we do agree that Tay et al., 2011 extended the results of Poliseno et al., 2010 to identify additional ceRNAs that modulate PTEN through microRNA competition and have included this in the revised manuscript.

Reviewer #3:[...] This Pandolfi group publication was seen as a major advancement on the field of non-codingRNAs at the time of publication (see the comments in Nature 2010 and Nat Rev Cancer 2010). This publication inspired also a large effort by Chinnaiyan's group who produced the translational landscape of expressed pseudogenes in human cancers. Therefore, the Poliseno et al., publication had, in fact, a major impact on the field of non-codingRNAs independent of Dr. Pandolfi's own research.This is the basis of the selection of the paper to be reproduced, making consequently the report by Kerwin and colleagues more interesting regarding data reproducibility for a large spectrum of readers independently on the specific focus of their research. It has to be appreciated the effort and work and time commitment by Kerwin and colleagues on this.1) Regarding the Abstract, this report is from a series of reports on the same theme and has the wording pattern from other similar reports published from the same group. What is surprisingly missing from all these abstracts is the conclusion sentence that is a standard for scientific publications. Please add a conclusion: did this report generally confirmed or not the data from Poliseno et al., 2010? I would recommend also for future studies on other publications to have a conclusive statement in the Abstract.

We've revised the abstract to include additional information, but we did not include a statement regarding whether this report generally confirmed (or not) the data from Poliseno et al., 2010, similar to our other reports. *eLife* has been publishing editors' summaries from their perspective of whether the results were generally confirmed (or not) and since there is no agreement of what it means to confirm (or not) an original study we prefer to remain silent on making this claim in the individual replication reports. Finally, we will publish a final paper that summarizes the collective evidence of all the replications, which will provide a better platform for discussing this important and complex issue.

2) In Figure 1, I would add a second panel with the actual levels of downregulation of PTEN, PTENP1 and PTEN/PTENP1, and also compare these levels with the ones reported by Poliseno et al. If these are not reported in Nature paper from 2010, then this information has to be requested from Pandolfi's group. It is essential, as various levels of downregulation can give various phenotype effects.

We agree that various levels of downregulation can give various phenotype effects and have included this in the revised manuscript. Specific to this point, the downregulation of PTEN, PTENP1, and PTEN/PTENP1 are included in Figure 2 and directly compared to what was reported in Poliseno et al., 2010 in the discussion and in Figure 6. We have revised the transition between Figure 1 and Figure 2 to make this point clearer for readers.

3) The lack of replication of reported data in subsection "Quantitative PCR following transfected with siRNA against PTEN and/or PTENP1", meaning the lack of significant variation of PTEN mRNA levels following PTENP1 knockdown and vice versa are problematic, as these are related to the core finding of Poliseno's et al. paper... Starting from simple technical points, were the Î² actin levels by qRT-PCR in the replication study comparable with the ones from Poliseno's study? If not, then this can be an explanation. Second, was the confluence of cells in the experiment by Kerwin and colleagues' for this particular experiment similar with the cell confluence levels used by Poliseno et al.? It is already reported that cell confluence significantly modifies the levels of expression of ncRNAs, see please a seminal paper by Mandel's group (Hwang et al., 2009). As this info is not reported in Nature paper and also is not used to be reported in publications, direct communication with the authors from Poliso et al., paper is important. Generally, as this is such an essential point for the ceRNA theory, the data have to be compared point by point between the two groups.

We did have direct contact with the original authors during the design phase of the project, which resulted in the Registered Report, but did not receive the data or information raised in this comment. We agree that these are important points to consider and have included these in the revised manuscript despite the inability to provide a point by point comparison of the levels.

4) For Figure 3B, please add the quantification results of PTEN protein levels. Same question as above: was the confluence of cells in the experiment by Kerwin and colleagues' fir this particular experiment similar with the cell confluence levels for Poliseno et al.?

The quantification of the PTEN protein levels, normalized to HSP90 are included in Figure 3A. We have revised the 3B to include the quantified values for this biological repeat for added clarity.

5) The lack of replication of reported data in subsection "Quantitative PCR following PTEN 3` UTR transfection", meaning the lack of effect of exogenous PTEN-3UTR on the expression of PTENP1 is also problematic regarding Poliseno's report...Again, the levels of normalizers for the two sets of experiments, Kerwin and colleagues versus Poliseno et al., as well as cell confluence, the cell passage used for these studies, have to be compared side by side. If not possible, I suggest the inclusion of this points specifically in the discussion, as a knowledgeable reader can't draw conclusions without such information.

We agree that these are important points to consider and have included these in the revised manuscript despite the inability to provide a point by point comparison of the levels.